# Towards Atoms of Large Language Models

Chenhui Hu [1 2]   Pengfei Cao [1 2]   Yubo Chen [1 2]   Kang Liu [1 2]   Jun Zhao [1 2]

## Abstract

The fundamental representational units (FRUs) of large language models (LLMs) remain undefined, limiting further understanding of their underlying mechanisms. In this paper, we introduce **Atom Theory** to systematically define, evaluate, and identify such FRUs, which we term atoms. Building on the atomic inner product (AIP), a non-Euclidean metric that captures the underlying geometry of LLM representations, we formally define atoms and propose two key criteria for ideal atoms: faithfulness ($R^2$) and stability ($q^*$). We further prove that atoms are identifiable under threshold-activated sparse autoencoders (TSAEs). Empirically, we uncover a pervasive representation shift in LLMs and demonstrate that the AIP corrects this shift to capture the underlying representational geometry. We find that two widely used units, neurons and features, fail to qualify as ideal atoms: neurons are faithful ($R^2 = 1$) but unstable ($q^* = 0.5\%$), while features are more stable ($q^* = 68.2\%$) but unfaithful ($R^2 = 48.8\%$). To find atoms of LLMs, leveraging atom identifiability under TSAEs, we show via large-scale experiments that reliable atom identification occurs only when the TSAE capacity matches the data scale. Guided by this insight, we identify FRUs with near-perfect faithfulness ($R^2 = 99.9\%$) and stability ($q^* = 99.8\%$) across layers of Gemma2-2B, Gemma2-9B, and Llama3.1-8B, satisfying the criteria of ideal atoms statistically. Further analysis confirms that these atoms align with theoretical expectations and exhibit substantially higher monosemanticity. Overall, we propose and validate Atom Theory as a foundation for understanding the internal representations of LLMs.

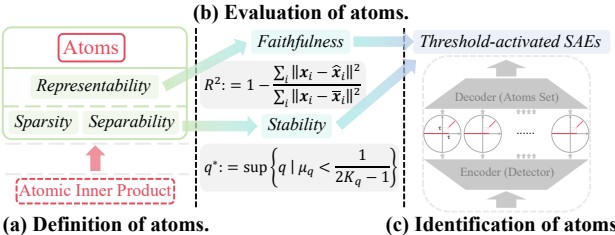

**(b) Evaluation of atoms.**

Atoms

*Representability*  Faithfulness  *Threshold-activated SAEs*

$$R^2 := 1 - \frac{\sum_i \|x_i - \hat{x}_i\|^2}{\sum_i \|x_i - \overline{x}_i\|^2}$$

Sparsity  Separability  Stability

Decoder (Atoms Set)

$$q^* := \sup\left\{q \mid \mu_q < \frac{1}{2K_q - 1}\right\}$$

Encoder (Detector)

Atomic Inner Product

**(a) Definition of atoms.**         **(c) Identification of atoms.**

*Figure 1.* Illustration of Atom Theory. (a) Atoms are defined based on the atomic inner product, inducing representability, sparsity, and separability. (b) Atoms are evaluated by faithfulness ($R^2$) and stability ($q^*$), measuring fidelity and stable-atom fraction. (c) Threshold-activated SAEs enable atom identification, with the encoder as an atom detector and the decoder as the target atom set.

## 1. Introduction

Large language models (LLMs), trained on vast corpora, exhibit emergent knowledge and reasoning abilities (Petroni et al., 2019; Brown et al., 2020; Achiam et al., 2023). Yet such information is not stored in explicit symbolic structures, but implicitly embedded within high-dimensional representations (Nanda et al., 2023; Gurnee et al., 2023). This raises a fundamental question: **Do LLMs contain fundamental representational units (FRUs)—an atomic structure underlying how they encode and compose information?** These representational units are critical for understanding, interpreting, and controlling LLMs (Olah et al., 2020).

Traditionally, neurons have been regarded as such FRUs (Olah et al., 2017; Dai et al., 2022; Chen et al., 2024). However, neurons frequently exhibit substantial polysemanticity (Elhage et al., 2022), raising doubts about their validity for analysis. To address this issue, features extracted from internal representations (Cunningham et al., 2023; Chen et al., 2025) have been proposed as alternative FRUs (Olah et al., 2020). Yet this perspective remains controversial: (i) features fail to fully reconstruct the original representations (Bricken et al., 2023), raising concerns about faithfulness; and (ii) features undergo splitting into finer ones or merging into broader ones under varying decomposition settings (Bussmann et al., 2025; Chanin et al., 2025), undermining stability. Although prior work implicitly treats neurons and features as FRUs, there is still no formal definition of FRUs for LLMs, which hinders principled evaluation and identification and ultimately limits theoretical clarity.

[1]The Key Laboratory of Cognition and Decision Intelligence for Complex Systems, Institute of Automation, Chinese Academy of Sciences, Beijing, China. [2]School of Artificial Intelligence, University of Chinese Academy of Sciences, Beijing, China. Correspondence to: Pengfei Cao <pengfei.cao@nlpr.ia.ac.cn>, Jun Zhao <jzhao@nlpr.ia.ac.cn>.

*Proceedings of the 43rd International Conference on Machine Learning*, Seoul, South Korea. PMLR 306, 2026. Copyright 2026 by the author(s).

In this paper, we propose *Atom Theory* to systematically define, evaluate, and identify the FRUs of LLMs, termed atoms (Fig. 1). Specifically, to characterize the underlying geometry of LLM representations, we introduce the (non-Euclidean) atomic inner product (AIP) in § 3.1. Based on AIP, we formally define atoms (Fig. 1(a)) in § 3.2 by three properties: **representability** (representations can be faithfully reconstructed from atoms), **sparsity** (each representation involves only a few atoms), and **separability** (atoms are approximately orthogonal under AIP). Representability ensures that atoms form a faithful set for the representation space. Sparsity and separability jointly enable efficient encoding of representations by approximately orthogonal atoms (Elhage et al., 2022). Accordingly, sparsity and separability are tightly coupled, and we explicitly quantify their relationship in the following criteria.

To operationalize this definition, we next introduce quantitative criteria for evaluating whether candidate units qualify as atoms (Fig. 1(b)), theoretically in § 3.3 and practically in § 4.2. Representability is measured by the coefficient of determination $R^2$, which quantifies *faithfulness*. Sparsity and separability, drawing on compressed sensing (Donoho, 2006; Candès et al., 2006), are unified into a single metric $q^*$ to quantify *stability*. This metric corresponds to monorepresentationality: within the regime characterized by $q^*$, atoms and their combinations are distinguishable (i.e., not confusable), which is a desirable property under approximate orthogonality. Monorepresentationality provides a structural foundation for understanding LLM representations, offering the stability required for monosemanticity. Finally, to provide theoretical guarantees for atom identification, we prove that threshold-activated sparse autoencoders (TSAEs) (Fig. 1(c)) can identify the target atom set. Overall, we establish a unified theoretical framework that provides guarantees for defining, evaluating, and identifying atoms.

**Having introduced *Atom Theory*, we validate that its foundation, the AIP, provides a principled basis for understanding LLM representations in § 4.1.** Empirically, we uncover a pervasive representation shift across layers of differently scaled models from multiple LLM families (Fig. 2), including GPT (Radford et al., 2019; Wang & Komatsuzaki, 2022), Pythia (Biderman et al., 2023), Llama (Touvron et al., 2023; Dubey et al., 2024), and Gemma (Team et al., 2024). This shift arises from the Softmax operation in LLMs, which drives the centroid of the distribution of pairwise representation angles away from $90°$ under the Euclidean inner product, thereby inducing a global bias in the representation space and distorting the representational geometry. Introducing the AIP effectively corrects this shift (Fig. 3), removing the global bias and restoring the centroid to $90°$. This demonstrates that the AIP captures the underlying geometry of LLM representations.

**Building on *Atom Theory*, we evaluate whether candidate representational units satisfy the criteria for ideal atoms in § 4.2.** We show that widely used representational units, neurons and features, remain distant from ideal atoms (Fig. 4). Although neurons, as the basic computational units of neural networks, exhibit perfect faithfulness ($R^2 = 1$), they display extremely low stability ($q^* = 0.5\%$). Features achieve improved stability ($q^* = 68.2\%$) but remain unstable and exhibit low faithfulness ($R^2 = 48.8\%$). These results quantitatively reveal the limitations of neurons and features, indicating that these common units are not ideal atoms.

**Leveraging *Atom Theory*, we identify the atoms of LLMs in § 4.3 and § 4.4.** Based on the theoretical identifiability of TSAEs, we conduct large-scale experiments to characterize the relationship between data scale and TSAE capacity (Fig. 5), showing that reliable atom identification is achieved only when the TSAE capacity exceeds a critical threshold for a given data scale. This is intuitive: data scale determines the scale of atoms, which in turn dictates the TSAE capacity required for their identification. Guided by this insight, we achieve faithful reconstructions ($R^2 = 99.90\%$) across layers of Gemma2-2B, Gemma2-9B, and Llama3.1-8B using TSAEs with JumpReLU activation (Erichson et al., 2019; Rajamanoharan et al., 2024b), and verify high stability of the identified units ($q^* = 99.85\%$), yielding ideal atoms statistically (Tab. 1). Further analysis demonstrates that the identified atoms are consistent with theoretical expectations and exhibit substantially higher monosemanticity (Fig. 6).

In summary, our contributions are as follows:

- We propose Atom Theory, a rigorous theoretical framework based on AIP that systematically defines, evaluates, and identifies the FRUs of LLMs, i.e., atoms.

- We empirically uncover a representation shift in LLMs and show that the AIP corrects this shift to characterize the underlying representational geometry, validating the representational foundation of Atom Theory.

- Building on Atom Theory, we use faithfulness and stability to systematically and quantitatively reveal the limitations of neurons and features as FRUs.

- Leveraging Atom Theory, we establish atom identifiability under TSAEs, characterize the relationship between data scale and TSAE capacity, and identify FRUs in LLMs that satisfy the criteria of ideal atoms.[1] Further analysis shows that these atoms align with theoretical expectations and exhibit high monosemanticity.

## 2. Preliminary

**Background on Language Models**    Consider an $L$-layer language model over a vocabulary $V$. For an input sequence

---

[1]Code available at https://github.com/ChenhuiHu/towards_atoms.

$\boldsymbol{x} = [x_1, x_2, \cdots, x_T]$ with $x_i \in V$, the model assigns each token $x_i$ an embedded representation $\boldsymbol{h}_i^0 \in \mathbb{R}^H$, which is updated at layer $l$ as $\boldsymbol{h}_i^l = \boldsymbol{h}_i^{l-1} + \boldsymbol{a}_i^l + \boldsymbol{v}_i^l$, where $\boldsymbol{a}_i^l$ and $\boldsymbol{v}_i^l$ denote the outputs of attention and MLP modules, respectively. From the residual-stream perspective, the representation after $L$ layers is $\boldsymbol{h}_i^L = \boldsymbol{h}_i^0 + \sum_{l=1}^L \boldsymbol{a}_i^l + \sum_{l=1}^L \boldsymbol{v}_i^l$. The probability distribution $\boldsymbol{y}$ over the next token is obtained from the final representation via $\boldsymbol{y} = \mathrm{Softmax}(\boldsymbol{W}_U^\top \boldsymbol{h}_T^L)$, where $\boldsymbol{W}_U \in \mathbb{R}^{H \times |V|}$ is the unembedding matrix.

## 3. Atom Theory

In language models, all information is embedded in high-dimensional representations. Our objective is to identify the fundamental representational units (FRUs), which we term atoms. Formally, we consider a collection of representations $M = \{\boldsymbol{m}_i\}_{i=1}^{|M|}$, where $\boldsymbol{m}_i \in \mathbb{R}^H$. Each representation can be expressed as $\boldsymbol{m}_i = \sum_j \delta(i,j) \boldsymbol{d}_j$, where $\delta(i,j) \geq 0$ denotes the presence and magnitude of the $j$-th representational unit $\boldsymbol{d}_j \in \mathbb{R}^H$ in the $i$-th representation $\boldsymbol{m}_i$. However, in representation space, the family of representational units admitting such a decomposition is, in principle, infinite.

The central question is how to define FRUs, i.e., atoms. A natural criterion is distinguishability: each atom should be detectable or manipulable without interfering with others. In high-dimensional spaces, this criterion translates into orthogonality: atoms occupy mutually orthogonal directions, making their identities distinguishable via inner products. This motivates introducing an inner product to analyze the geometry of FRUs. Thus, the choice of inner product is critical. While the Euclidean inner product is commonly used, it is not necessarily appropriate for language models. Following Park et al. (2023) and Hu et al. (2025), we consider the following reparameterization of $\boldsymbol{W}_U$ and $\boldsymbol{h}^L$:

$$\boldsymbol{W}_U{}' \leftarrow \boldsymbol{A}^{-\top} \boldsymbol{W}_U + \boldsymbol{b}\,\boldsymbol{1}^\top, \quad \boldsymbol{h}'^L \leftarrow \boldsymbol{A}\,\boldsymbol{h}^L, \qquad (3.1)$$

where $\boldsymbol{A} \in \mathbb{R}^{H \times H}$ is an invertible linear transform, $\boldsymbol{b} \in \mathbb{R}^H$, and $\boldsymbol{1} \in \mathbb{R}^{|V|}$ is the all-ones vector. Owing to the translation invariance of Softmax, this reparameterization leaves the output distribution unchanged: $\boldsymbol{y} = \mathrm{Softmax}(\boldsymbol{W}_U^\top \boldsymbol{h}^L) = \mathrm{Softmax}(\boldsymbol{W}_U'^\top \boldsymbol{h}'^L)$. See Appendix A.1 for further details.

Since the training objective of language models depends on representations solely through Softmax probabilities, different pairs $(\boldsymbol{W}_U, \boldsymbol{h}^L)$ under (3.1) are observationally indistinguishable, producing exactly the same outputs for all inputs. Thus, even for a trained checkpoint, such reparameterizations leave the model's input–output behavior unchanged, so $\boldsymbol{h}^L$ is identifiable only up to an invertible linear transformation $\boldsymbol{A}$ in principle. Due to the residual-stream architecture and linearity of matrix multiplication, this invariance propagates to all hidden representations and their representational units $\boldsymbol{d}_j$, which are likewise identifiable only up to $\boldsymbol{A}$.

Crucially, the Euclidean inner product is not invariant under reparameterization (3.1): $\langle \boldsymbol{d}_i, \boldsymbol{d}_j \rangle \neq \langle \boldsymbol{A}\boldsymbol{d}_i, \boldsymbol{A}\boldsymbol{d}_j \rangle$. Therefore, the Euclidean geometric relations (e.g., angles and orthogonality) between $\boldsymbol{d}_i$ and $\boldsymbol{d}_j$ depend on the chosen parameterization, thus the Euclidean inner product does not provide a canonical geometry for language-model representations.

### 3.1. Atomic Inner Product

To better understand language-model representations and thereby define atoms within them, we require additional principles to specify an appropriate inner product. We therefore introduce an inner product with the desired properties.

**Definition 3.1** (Atomic Inner Product; AIP). Let $\mathcal{D} = \mathrm{span}(D)$, where $D = \{\boldsymbol{d}_j\}_{j=1}^{|D|}$ denotes the atom set. The **atomic inner product** $\langle \cdot, \cdot \rangle_S$ is an inner product on $\mathcal{D}$ such that $\langle \boldsymbol{d}_i, \boldsymbol{d}_j \rangle_S = 0$ for all $\boldsymbol{d}_i, \boldsymbol{d}_j \in D$ with $i \neq j$.

Atoms are indexed arbitrarily, and any permutation of indices leaves their geometry invariant. Hence, there is no principled basis for assigning different norms to different atoms. By this symmetry, we assume a common norm under the chosen inner product, i.e., $\|\boldsymbol{d}_i\|_S = c > 0, \forall i \in [|D|]$. The constant $c$ cancels naturally in the subsequent analysis.

By abuse of notation, we also use $\boldsymbol{D} \in \mathbb{R}^{H \times |D|}$ to denote the matrix with columns $\boldsymbol{d}_j$. We next characterize the atomic inner product in an explicit form.

**Theorem 3.2** (Explicit Form of the Atomic Inner Product). *Let $\langle \boldsymbol{d}_i, \boldsymbol{d}_j \rangle_S = \boldsymbol{d}_i^\top \boldsymbol{S} \boldsymbol{d}_j$ be an atomic inner product with $\boldsymbol{S} \in \mathbb{R}^{H \times H}$ symmetric and positive definite. If the columns of $\boldsymbol{D} = [\boldsymbol{d}_1, \boldsymbol{d}_2, \cdots, \boldsymbol{d}_{|D|}]$ form the atom set such that $\forall i, \|\boldsymbol{d}_i\|_S = c > 0$, and $\mathcal{D} \simeq \mathbb{R}^H$, then $\boldsymbol{S} = c^2 (\boldsymbol{D}\boldsymbol{D}^\top)^{-1}$.*

All proofs are provided in Appendix A. By analogy with cosine similarity, we introduce the normalized atomic inner product to remove the dependence on $c$.

**Corollary 3.3** (Normalized Atomic Inner Product; NAIP). *Let the atomic inner product be defined by $\langle \boldsymbol{d}_i, \boldsymbol{d}_j \rangle_S = \boldsymbol{d}_i^\top \boldsymbol{S} \boldsymbol{d}_j$, where $\boldsymbol{S}$ is symmetric and positive definite. Suppose the columns of $\boldsymbol{D} = [\boldsymbol{d}_1, \cdots, \boldsymbol{d}_{|D|}]$ form the atom set satisfying $\forall i, \|\boldsymbol{d}_i\|_S = c > 0$. Then, for any $i, j$,*

$$\langle \boldsymbol{d}_i, \boldsymbol{d}_j \rangle_{\tilde{S}} := \frac{\langle \boldsymbol{d}_i, \boldsymbol{d}_j \rangle_S}{\|\boldsymbol{d}_i\|_S \|\boldsymbol{d}_j\|_S} = \boldsymbol{d}_i^\top \tilde{\boldsymbol{S}} \boldsymbol{d}_j, \quad \tilde{\boldsymbol{S}} = (\boldsymbol{D}\boldsymbol{D}^\top)^{-1}.$$
$$(3.2)$$

*Consequently, the bilinear form $\langle \boldsymbol{d}_i, \boldsymbol{d}_j \rangle_{\tilde{S}} = \boldsymbol{d}_i^\top \tilde{\boldsymbol{S}} \boldsymbol{d}_j$ defines the **normalized atomic inner product**.*

Unlike the causal inner product (Park et al., 2023), which is defined in the static and input-independent unembedding space over output tokens, our AIP (or NAIP) operates on the dynamic and input-dependent representation space, characterizing the geometry of representations and their constituent units. This enables direct analysis of internal representations

and thereby allows the definition of atoms as fundamental representational units under the AIP.

*Remark.* Define $\tilde{d}_i = \tilde{S}^{\frac{1}{2}} d_i$ and $\tilde{d}_j = \tilde{S}^{\frac{1}{2}} d_j$. Under this transformation, $\langle d_i, d_j \rangle_{\tilde{S}} = \langle \tilde{d}_i, \tilde{d}_j \rangle$, where the right-hand side denotes the Euclidean inner product. Hence, properties of the Euclidean inner product transfer directly to the NAIP; $\tilde{d}_i$ and $\tilde{d}_j$ are accordingly termed ***normalized atoms***.

### 3.2. Formal Definition of Atoms

Having established a principled perspective for understanding representations in language models, we proceed to formally define atoms. However, the preceding analysis assumes an idealized setting in which atoms are strictly orthogonal under the chosen inner product. Although this assumption yields a clean notion, it constrains the number of atoms to at most the representation dimension ($|D| = H$), rendering the formulation impractical.

Elhage et al. (2022) observed that sparsity induces the emergence of approximately orthogonal representations in neural networks to cope with limited representational dimensionality, a phenomenon termed superposition. Hu et al. (2025) later identified pervasive superposition in language models. Motivated by these findings, we introduce sparsity, which naturally leads to approximate orthogonality and enables a formally grounded, practical definition of atoms.

**Definition 3.4** (Sparsity Level)**.** Let $M = \{m_i\}_{i=1}^{|M|} \subset \mathbb{R}^H$ be a collection of representations. Suppose there exist $D = [d_1, \cdots, d_{|D|}] \in \mathbb{R}^{H \times |D|}$ and $\delta_i \in \mathbb{R}_{\geq 0}^{|D|}$ such that $m_i = D\delta_i$, $\forall i \in [|M|]$. The **sparsity level** is $K := \max_i \|\delta_i\|_0$, where $\|\cdot\|_0$ denotes the $\ell_0$ norm.

Sparsity enables the number of atoms to substantially exceed the ambient dimension, yielding an overcomplete structure with $|D| \gg H$ that captures rich world knowledge, while simultaneously minimizing mutual interference.

*Remark.* In the basic setting of § 3.1 with $|D| = H$, one can verify that $\tilde{S} = (DD^\top)^{-1}$ satisfies $D^\top \tilde{S} D = I_{|D|}$, so the atoms form an orthonormal basis under $\langle \cdot, \cdot \rangle_S$. When $|D| \gg H$, $\tilde{S} = (DD^\top)^{-1}$ is well defined provided $\text{rank}(D) = H$, but $G := D^\top \tilde{S} D$ becomes a rank-$H$ projection rather than $I_{|D|}$, thus the atoms cannot all be mutually orthogonal under $\langle \cdot, \cdot \rangle_S$. Therefore, exact orthogonality is unattainable, motivating the introduction of approximate orthogonality.

This consideration motivates the following definition.

**Definition 3.5** ($\epsilon$-Approximately Orthogonal Atoms)**.** The atom set $\{d_i\}_{i=1}^{|D|}$ is $\epsilon$-**approximately orthogonal** if $|\langle d_i, d_j \rangle_{\tilde{S}}| \leq \epsilon$, $\forall i \neq j$, where $\langle x, y \rangle_{\tilde{S}} := x^\top \tilde{S} y$ denotes the normalized atomic inner product and $\tilde{S} := (DD^\top)^{-1}$.

*Remark.* In the ideal setting of exact orthogonality, the inner products $\langle d_i, d_j \rangle_{\tilde{S}}$ (equivalently, $\langle \tilde{d}_i, \tilde{d}_j \rangle$) for $i \neq j$ follow a Dirac measure concentrated at the origin. Under

$\epsilon$-approximate orthogonality, they are instead expected to follow a Gaussian distribution $\mathcal{N}(0, s^2)$ with small variance, which converges to the Dirac measure as $s \to 0$.

We now introduce a formal definition of atoms.

**Definition 3.6** (Atoms)**.** Let $M = \{m_i\}_{i=1}^{|M|}$ be a collection of representations. Suppose there exists $D = [d_1, \cdots, d_{|D|}] \in \mathbb{R}^{H \times |D|}$ and $\Delta = [\delta_1, \cdots, \delta_{|M|}] \in \mathbb{R}_{\geq 0}^{|D| \times |M|}$ such that, for a given sparsity level $K \in \mathbb{N}$,

$$\forall i \in [|M|], \quad m_i = D\delta_i \text{ with } \|\delta_i\|_0 \leq K. \quad (3.3)$$

Furthermore, $|\langle d_i, d_j \rangle_{\tilde{S}}| \leq \epsilon$, $\forall i \neq j$, where $\tilde{S} := (DD^\top)^{-1}$. Under these conditions, $\{d_i\}_{i=1}^{|D|}$ is called the **atom set** of $M$, and each $d_i$ is referred to as an **atom**.

Intuitively, atoms are characterized by three properties: **representability**, where each representation can be faithfully expressed by atoms, i.e., $m_i = D\delta_i$; **sparsity**, where each representation involves only a few atoms, i.e., $\|\delta_i\|_0 \leq K$; and **separability**, where atoms are approximately orthogonal, i.e., $|\langle \tilde{d}_i, \tilde{d}_j \rangle| \leq \epsilon$. Representability is a natural requirement, while sparsity and separability jointly enable efficient encoding of separable information under approximate orthogonality. We further quantitatively characterize the relationship between sparsity and separability in § 3.3.

*Remark.* $K$ quantifies sparsity without enforcing a specific sparsity regime, ensuring broad applicability of the definition. Moreover, pre-multiplying both sides of (3.3) by $\tilde{S}^{\frac{1}{2}}$ yields $\tilde{m}_i = \tilde{D}\delta_i$, with $\tilde{m}_i := \tilde{S}^{\frac{1}{2}} m_i$ and $\tilde{D} := \tilde{S}^{\frac{1}{2}} D = [\tilde{d}_1, \cdots, \tilde{d}_{|D|}]$, which simplifies subsequent derivations.

### 3.3. Evaluation of Atoms

Having formalized atoms via representability, sparsity, and separability, we now evaluate whether candidate representational units satisfy the criteria of ideal atoms. Representability is measured by the coefficient of determination $R^2 := 1 - \frac{\sum_i \|x_i - \hat{x}_i\|^2}{\sum_i \|x_i - \bar{x}\|^2}$, which quantifies the proportion of variance in the original representations explained by atoms and thus reflects faithfulness. While sparsity and separability can be quantified by $K$ and $\epsilon$, they do not directly indicate unit quality. In this section, we introduce a unified metric that integrates sparsity and separability to characterize the stability of representational units.

We note that $\delta_i \in \mathbb{R}^{|D|}$ can be viewed as a sparse representation of high-dimensional semantics, which is compressed by $D \in \mathbb{R}^{H \times |D|}$ to produce $m_i \in \mathbb{R}^H$. This perspective reveals a close connection to compressed sensing (Donoho (2006); Candès et al. (2006)), whose core idea is that a signal sparse in some basis can be recovered from far fewer linear measurements than its ambient dimension, with the Restricted Isometry Property (RIP) (Goeßmann & Kutyniok, 2020; Chen & Schaeffer, 2021) providing the essential guarantee.

**Definition 3.7** (Restricted Isometry Property; RIP). *A matrix $\tilde{D} \in \mathbb{R}^{H \times |D|}$ is said to satisfy the $K$-RIP if there exists a constant $\delta_K \in [0, 1)$ such that, for any $K$-sparse vector $\boldsymbol{\delta} \in \mathbb{R}^{|D|}$ (i.e., $\|\boldsymbol{\delta}\|_0 \leq K$),*

$$(1 - \delta_K)\|\boldsymbol{\delta}\|_2^2 \leq \|\tilde{D}\boldsymbol{\delta}\|_2^2 \leq (1 + \delta_K)\|\boldsymbol{\delta}\|_2^2. \quad (3.4)$$

*Here, $\delta_K$ is called the $K$-RIP constant of $\tilde{D}$.*

Intuitively, projecting a sparse vector via $\tilde{D}$ to lower dimension preserves its geometric structure, ensuring the possibility of recovery. Direct verification of the RIP is NP-hard, but coherence (Foucart & Rauhut, 2013; Murdock & Lucey, 2020) provides a computable upper bound on $\delta_k$.

**Theorem 3.8** (Coherence–RIP Upper Bound). *Let $\tilde{D} \in \mathbb{R}^{H \times |D|}$ and define the coherence $\mu := \max_{i \neq j} |\langle \tilde{d}_i, \tilde{d}_j \rangle| \leq \varepsilon$. For any $K$-sparse vector $\boldsymbol{\delta} \in \mathbb{R}^{|D|}$ with $\|\boldsymbol{\delta}\|_0 \leq K$,*

$$\big(1 - (K-1)\mu\big)\|\boldsymbol{\delta}\|_2^2 \leq \|\tilde{D}\boldsymbol{\delta}\|_2^2 \leq \big(1 + (K-1)\mu\big)\|\boldsymbol{\delta}\|_2^2. \quad (3.5)$$

*Hence $\delta_K(\tilde{D}) \leq (K-1)\mu$; in particular, $\tilde{D}$ satisfies the $K$-RIP whenever $(K-1)\mu < 1$.*

In other words, coherence provides a computable criterion for verifying the RIP, ensuring that all $K$-sparse vectors projected through $\tilde{D}$ preserve geometric structure, an essential prerequisite in compressed sensing. Nevertheless, the RIP alone does not preclude non-uniqueness (Donoho et al., 2001; Candes & Tao, 2005): even if $(K-1)\mu < 1$ holds, the sparse coefficients associated with representations need not be unique.

**Theorem 3.9** (Uniqueness and Exact $\ell_1$ Recoverability). *Let $\tilde{D} \in \mathbb{R}^{H \times |D|}$ and define the coherence $\mu := \max_{i \neq j} |\langle \tilde{d}_i, \tilde{d}_j \rangle| \leq \varepsilon$. If $\mu < \frac{1}{2K-1}$, then for every $\boldsymbol{\delta} \in \mathbb{R}^{|D|}$ with $\|\boldsymbol{\delta}\|_0 \leq K$, the $K$-sparse representation determined by $\tilde{m} = \tilde{D}\boldsymbol{\delta}$ is unique; that is, no other $K$-sparse vector yields the same $\tilde{m}$. Moreover, $\boldsymbol{\delta}$ is the unique minimizer of the convex program*

$$\min_{\boldsymbol{x} \in \mathbb{R}^{|D|}} \|\boldsymbol{x}\|_1 \quad subject \ to \quad \tilde{D}\boldsymbol{x} = \tilde{m}. \quad (3.6)$$

Crucially, this intrinsically characterizes the monorepresentationality of $\tilde{D}$: under the condition $\mu < \frac{1}{2K-1}$, any representation formed as a $K$-sparse linear combination of representational units in $\tilde{D}$ has a unique sparse decomposition, with no other combination yielding the same representation.

**Corollary 3.10** (Monorepresentationality / Injectivity). *Under the condition $\mu < \frac{1}{2K-1}$ of Theorem 3.9, define $\Sigma_K := \{\boldsymbol{\delta} \in \mathbb{R}^{|D|} : \|\boldsymbol{\delta}\|_0 \leq K\}$ and $\Phi : \Sigma_K \to \mathbb{R}^H$ by $\Phi(\boldsymbol{\delta}) := \tilde{D}\boldsymbol{\delta}$. Then $\Phi$ is injective on $\Sigma_K$. That is, for any $\boldsymbol{x}, \boldsymbol{y} \in \Sigma_K$, if $\tilde{D}\boldsymbol{x} = \tilde{D}\boldsymbol{y}$, it follows that $\boldsymbol{x} = \boldsymbol{y}$.*

Within this regime, representational units and their combinations are unambiguous in representation space, i.e.,

monorepresentationality. By contrast, monosemanticity (Bricken et al. (2023); Templeton (2024)) concerns the alignment of a unit with a specific meaning, concept, or function. The former is formally provable, whereas the latter is statistical and interpretive. Monorepresentationality is a prerequisite for monosemanticity, as it provides the structural stability; otherwise, units would be non-unique and interpretation unstable. Without additional semantic anchoring or inductive assumptions, theoretical guarantees primarily concern structural stability induced by monorepresentationality, while monosemanticity must be determined empirically via further semantic alignment experiments.

Sparsity ($K$) and separability ($\mu$) are thus unified by the condition $\mu < \frac{1}{2K-1}$, under which stability holds. More practical metrics are discussed in § 4.2 and Appendix C.5.

*Remark.* For intuition, the above result can be viewed as a generalization of the strictly orthogonal case ($|D| = H$). In this setting, $\tilde{D} \in \mathbb{R}^{H \times H}$ has orthogonal columns ($\mu = 0$), i.e., $\tilde{D}^\top \tilde{D} = I$. Any representation $\tilde{m} \in \mathbb{R}^H$ then admits a unique decomposition $\boldsymbol{\delta} = \tilde{D}^\top \tilde{m}$, satisfying $\tilde{m} = \tilde{D}\boldsymbol{\delta}$. Here the sparsity level $K = H = |D|$ satisfies $\mu < \frac{1}{2K-1}$. Thus, under strict orthogonality, representations and atom coefficients are in one-to-one correspondence, yielding a direct and unique identification of atoms.

### 3.4. Identification of Atoms

We have defined atoms and established criteria for ideal atoms. A central question remains whether such atoms can be identified in practice. Since sparse autoencoders (SAEs) are a standard approach for learning disentangled representations (Cunningham et al., 2023), we next demonstrate that, under appropriate conditions, SAEs can indeed identify these atoms, rendering the theory practically applicable.

**Theorem 3.11** (Identifiability of Threshold-activated SAEs; TSAEs). *Let $M = \{m_i\}_{i=1}^{|M|} \subset \mathbb{R}^H$ with $m_i = D\boldsymbol{\delta}_i$, where $D = [d_1, \cdots, d_{|D|}] \in \mathbb{R}^{H \times |D|}$ satisfies $|\langle \tilde{d}_i, \tilde{d}_j \rangle| \leq \epsilon$ for all $i \neq j$. Suppose each $\boldsymbol{\delta}_i \in \mathbb{R}^{|D|}$ is $K$-sparse, i.e. $\|\boldsymbol{\delta}_i\|_0 \leq K$. Consider the threshold activation function*

$$\sigma_\tau(x) = \begin{cases} 0 & x < \tau, \\ x & x \geq \tau, \end{cases} \quad (3.7)$$

*with threshold $\tau > 0$. Assume there exist constants $0 < \delta_{\min} \leq \delta_{\max}$ such that, for each support $\mathcal{S}_i = \text{supp}(\boldsymbol{\delta}_i)$, $\delta_{\min} \leq \delta_{ij} \leq \delta_{\max}$, $\forall j \in \mathcal{S}_i$. If the amplitude gap and threshold satisfy $\varepsilon K \delta_{\max} < \tau < \delta_{\min} - \varepsilon(K-1)\delta_{\max}$, which is feasible whenever $\delta_{\min} > \varepsilon(2K-1)\delta_{\max}$, then setting TSAEs with $W_{dec} = D$ and $W_{enc} = D^\top \tilde{S}$ yields*

$$\forall i, \quad W_{dec} \, \sigma_\tau(W_{enc} m_i) = m_i. \quad (3.8)$$

*This parameterization exactly recover the atom set $D$.*

Therefore, TSAEs can identify atoms in principle. By contrast, ReLU (Templeton, 2024), lacking a threshold term,

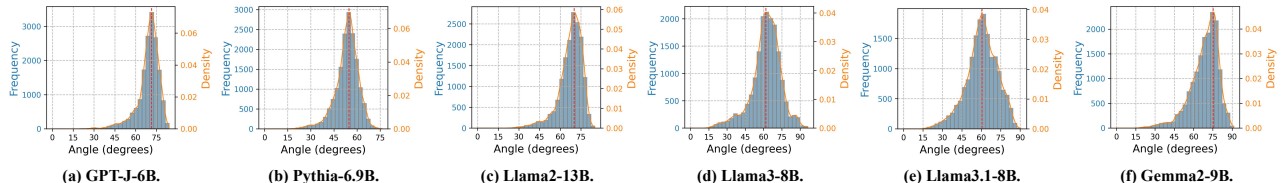

*Figure 2.* Representation shift at the final layer across multiple LLMs under the Euclidean inner product, with the centroid of pairwise representation angles deviating from $90°$. See Appendix B for full results.

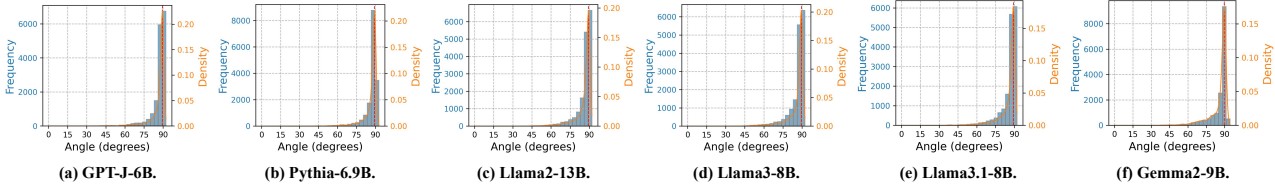

*Figure 3.* Correction of representation shift at the final layer across multiple LLMs via the atomic inner product, with the centroid of pairwise representation angles consistently approaching $90°$. See Appendix B for full results.

fails to satisfy the support-separation condition and is thus theoretically insufficient. This responds to O'Neill et al. (2024): the limitation of SAEs arises not from their linear–nonlinear mechanism, but from the absence of threshold activation, which prevents effective atom identification.

The threshold activation considered here denotes a class of activation functions rather than a specific instantiation. Existing examples include JumpReLU (Erichson et al., 2019; Rajamanoharan et al., 2024a) and Top$K$ (Makhzani & Frey, 2013; Gao et al., 2024). Although Top$K$ is analogous in some respects, it relies on a fixed $K$, limiting adaptivity and practical applicability. Moreover, the motivation differs fundamentally: threshold activation is proposed to enable effective atom identification under approximate orthogonality, whereas JumpReLU is introduced to address feature shrinkage (Rajamanoharan et al., 2024a), and Top$K$ is designed to directly control sparsity (Gao et al., 2024).

*Remark.* Although the theorem is stated for a uniform scalar threshold $\tau$, it extends directly to a coordinate-wise threshold vector $\boldsymbol{\tau}$, without affecting the squeeze condition or the proof. This generalization enlarges the feasible interval when activation magnitudes differ, thereby improving support separation and the robustness of atom identification.

## 4. Experiments

We next empirically validate and apply Atom Theory. In § 4.1, we uncover a pervasive representation shift in large language models (LLMs) and show that the atomic inner product (AIP) corrects it, capturing the representational geometry and validating the foundation of Atom Theory. In § 4.2, we use faithfulness and stability to quantitatively reveal the limitations of neurons and features as fundamental representational units (FRUs). Leveraging the identifiability guarantees of threshold-activated SAEs (TSAEs), we estab-

lish in § 4.3 the relationship between data scale and TSAE capacity through large-scale experiments. Finally, in § 4.4, we identify FRUs across LLMs that exhibit high faithfulness and stability, and demonstrate strong monosemanticity.

### 4.1. Representation Shift

In this section, we uncover a pervasive representation shift in LLMs and show that the AIP corrects it, capturing the representational geometry and grounding Atom Theory.

**Experimental Setup** We randomly sample 128 subject entities from WikiData (Vrandečić & Krötzsch, 2014) and extract the corresponding activations across all layers of multiple LLM families, including GPT-2, GPT-J, Pythia, Llama-2, Llama-3, Llama-3.1, and Gemma2, which serve as the target representations. This sample size suffices to characterize the distribution of pairwise representation angles, and additional samples do not change the overall distribution (see Appendix B, Figs. 10-11). It also facilitates ensuring no overlap with activations used in subsequent sampling.

We then collect 100K activations $\boldsymbol{k}$ per layer from Wikipedia corpora (with no overlap with the target representations) and compute $\mathbb{E}[\boldsymbol{k}\boldsymbol{k}^\top]$ to estimate the normalized AIP.

To analyze the distribution of representations, we use cosine similarity, which removes scale effects and aligns with the theoretical framework. For representations $\boldsymbol{u}, \boldsymbol{v} \in \mathbb{R}^H$, the cosine similarity under the Euclidean inner product (EIP) is defined as $\cos(\boldsymbol{u}, \boldsymbol{v}) = \frac{\langle \boldsymbol{u}, \boldsymbol{v} \rangle}{\|\boldsymbol{u}\|_2 \|\boldsymbol{v}\|_2}$. Under the AIP induced by $\tilde{\boldsymbol{S}}$, the corresponding cosine similarity is $\cos_{\tilde{S}}(\boldsymbol{u}, \boldsymbol{v}) = \frac{\langle \boldsymbol{u}, \boldsymbol{v} \rangle_{\tilde{S}}}{\|\boldsymbol{u}\|_{\tilde{S}} \|\boldsymbol{v}\|_{\tilde{S}}} = \frac{\boldsymbol{u}^\top \tilde{\boldsymbol{S}} \boldsymbol{v}}{\sqrt{\boldsymbol{u}^\top \tilde{\boldsymbol{S}} \boldsymbol{u}} \sqrt{\boldsymbol{v}^\top \tilde{\boldsymbol{S}} \boldsymbol{v}}}$, where $\tilde{\boldsymbol{S}}$ is estimated in practice as $(\mathbb{E}[\boldsymbol{k}\boldsymbol{k}^\top])^{-1}$. For clarity, cosine similarities are further converted into angles. See Appendix B for more details.

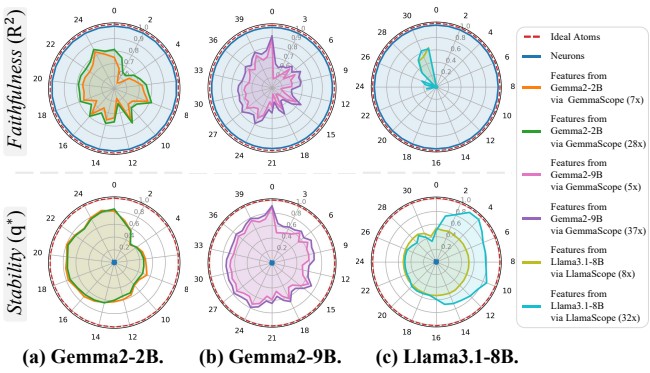

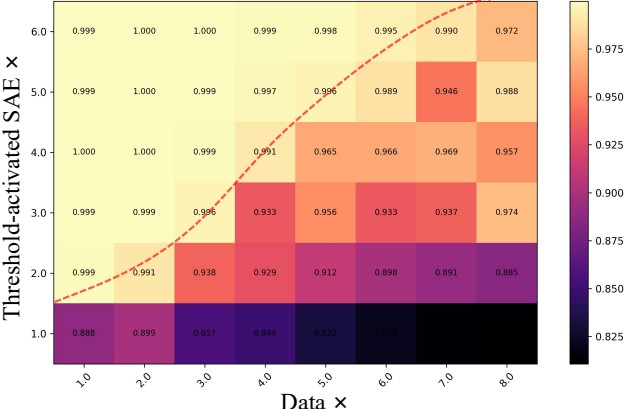

*Figure 4.* Comparison of neurons, features, and ideal atoms across all layers of different LLMs. Ideal atoms are required to exhibit both high faithfulness and high stability, corresponding to $R^2 = 1$ and $q^* = 1$, respectively. Values of $R^2$ below 0 are clipped to 0.

*Figure 5.* Matching TSAE capacity and data scale on Gemma2-2B (measured by $R^2$). Data × and TSAE × denote data scale and model capacity (interval 9,216). Red dashed lines mark the capacity range enabling reliable atom identification.

**Experimental Results** When representation angles are computed under the EIP, the centroid of the angular distribution deviates markedly from $90°$, revealing a representation shift (Fig. 2; full results in Appendix B, Figs. 7-9). This phenomenon consistently appears across all layers of LLM families, regardless of architectures or training corpora, indicating a systematic angular bias among representations. This bias shows that LLM representations are globally attracted toward a dominant direction, such that even unrelated representations exhibit high cosine similarity. As shown in Appendix B (Figs. 10–11), increasing the sample size used to characterize the distribution leaves the centroid unchanged, confirming attraction toward the same direction.

In contrast, the AIP corrects this shift, restoring the centroid of the angular distribution to $90°$ (Fig. 3; full results in Appendix B, Figs. 12-14). This indicates that the AIP removes the systematic angular bias induced by the EIP, so that angles between representations reflect genuine differences rather than metric-induced artifacts, which is essential for further analysis of LLM representations. Collectively, these results show that the AIP captures the underlying geometry of LLM representations and grounds Atom Theory.

### 4.2. Neurons or Features as Ideal Atoms?

In this section, we evaluate whether commonly used representational units, including neurons and features, satisfy the criteria of ideal atoms, i.e., faithfulness and stability.

**Experimental Setup** We use all 20K subject entities from the CounterFact dataset (Meng et al., 2022), a subset of WikiData, and extract the corresponding neurons and features (details in Appendix C.4) activated on these entities across all layers of Gemma2-2B, Gemma2-9B, and Llama3.1-8B as the target representational units.

Under Atom Theory, ideal atoms are evaluated along two dimensions. **Faithfulness** measures how accurately representational units reconstruct the original representations

and is quantified by the coefficient of determination $R^2 := 1 - \frac{\sum_i \|\boldsymbol{x}_i - \hat{\boldsymbol{x}}_i\|^2}{\sum_i \|\boldsymbol{x}_i - \bar{\boldsymbol{x}}\|^2}$, where $\hat{\boldsymbol{x}}_i$ and $\bar{\boldsymbol{x}}$ denote the predicted representation and sample mean, respectively. **Stability** captures structural robustness and is quantified by the maximal quantile $q^* := \sup\left\{ q \mid \mu_q < \frac{1}{2K_q - 1} \right\}$, where $\mu_q$ and $K_q$ denote quantile coherence and quantile sparsity. Ideal atoms satisfy $R^2 = 1$ and $q^* = 1$. Full definitions and evaluation details are provided in Appendix C.5.

**Experimental Results** As shown in Fig. 4, evaluation is summarized by two radar plots for faithfulness ($R^2$, upper panel) and stability ($q^*$, lower panel), whose axes correspond to layers of LLMs. Ideal atoms should approach 1 on each axis in both plots, thereby filling the unit circle in each plot. However, neurons, as the basic computational units of neural networks, exhibit perfect faithfulness ($\overline{R^2} = 1$) but extremely low stability ($\overline{q^*} = 0.5\%$), where $\bar{\cdot}$ denotes average over layers (and models). Features improve stability ($\overline{q^*} = 68.2\%$) relative to neurons, yet remain unstable and display low faithfulness ($\overline{R^2} = 48.8\%$). Theoretically, stability is necessary for monosemanticity (§ 3.3). Accordingly, the low stability of neurons implies polysemanticity, consistent with prior findings (Olah et al., 2020). Features exhibit higher stability and thus improved monosemanticity (Chen et al., 2025), but still fall short of ideal atoms. Overall, both neurons and features exhibit a clear gap from ideal atoms.

### 4.3. TSAE Capacity Meet Data Scale

Next, we address the practical problem of identifying the FRUs of LLMs. Theorem 3.11 shows that TSAEs are in principle capable of identifying atoms, but it does not specify how to choose the model capacity given the data scale in practice. In particular, we must determine the relationship between the data scale $|M|$ and the model capacity $|D|$ of TSAEs in order to reliably identify atoms.

*Table 1.* Faithfulness and stability of identified units across models. Reported values are averaged over all layers. More detailed and supplementary results are provided in Appendix C.5 (Tabs. 3–5).

| Model | *Faithfulness* ($R^2$) | *Stability* ($q^*$) |
|---|---|---|
| GEMMA2-2B | 99.92% | 99.74% |
| GEMMA2-9B | 99.93% | 99.87% |
| LLAMA3.1-8B | 99.85% | 99.95% |

**Experimental Setup** We use subject entities from Wiki-Data and extract the corresponding activations from the first layer of Gemma2-2B, yielding a sufficiently large representation dataset (1.9B samples), which is randomly sampled during training (details in Appendix C). Using an interval of 9,216, we evaluate $R^2$ obtained by training TSAEs across varying data scales and model capacities. Due to the high computational cost of this grid search, experiments are conducted exclusively on Gemma2-2B. As implied by the feasible interval condition in Theorem 3.11, faithfulness is achievable only when stability holds. For brevity, we report faithfulness here, and present extensive empirical results on stability of the identified atoms in § 4.4, confirming that the representations indeed exhibit a stable atomic structure.

**Experimental Results** As shown in Fig. 5, we find that high faithfulness occurs only when TSAE capacity exceeds a critical threshold for a given data scale. Intuitively, the data scale determines the scale of FRUs, which in turn determines required TSAE capacity for their identification. This finding also prompts a rethink of current SAE training paradigms, which heuristically choose model capacity and then train on massive activations from large corpora. While broadly applicable across tasks, their limited faithfulness ultimately constrains downstream reliability.

### 4.4. Atoms of LLMs

In this section, we identify widespread atoms of LLMs that exhibit high faithfulness and stability. Further analysis confirms that these atoms align with theoretical expectations and exhibit substantially stronger monosemanticity.

**Experimental Setup** Consistent with § 4.2, we use all subject entities from CounterFact and extract the corresponding activations across all layers of Gemma2-2B, Gemma2-9B, and Llama3.1-8B. Guided by the insights from § 4.3, we uniformly adopt a $4\times$ TSAE with JumpReLU activation (Erichson et al., 2019; Rajamanoharan et al., 2024b) to identify atoms. We evaluate the identified units with faithfulness ($R^2$) and stability ($q^*$). Full details on data, training, and computational costs are provided in Appendix C.

**Experimental Results** Across Gemma2-2B (layers 1–26), Gemma2-9B (layers 1–42), and Llama3.1-8B (layers 1–30), we consistently achieve high faithfulness ($\overline{R^2} = 99.90\%$) and stability ($\overline{q^*} = 99.85\%$), as shown in Tab. 1. These results indicate that the identified units approach ideal atoms

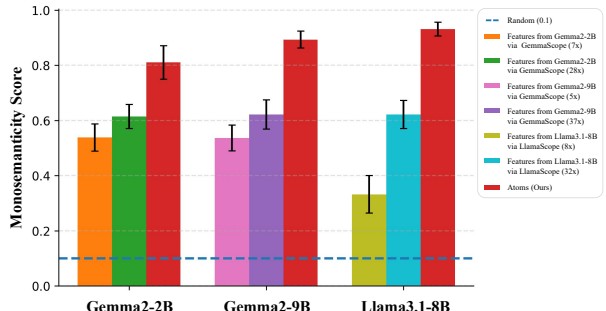

*Figure 6.* Monosemanticity scores of representational units across models, using GPT-5.2 with manual verification. The blue dashed line indicates random-guess performance (0.1).

statistically. Further analysis (see Appendix C.6) reveals that the training process is minimally sensitive to hyperparameters (Fig. 19); the encoder and decoder, when randomly initialized and trained independently without weight tying or additional constraints, converge to structures consistent with Theorem 3.11 (Fig. 20); the identified atoms are approximately orthogonal under AIP, approaching the theoretical Dirac distribution (Fig. 21).

Although stability provides a necessary structural foundation for monosemanticity, we validate this empirically. Specifically, we uniformly sample representational units across layers of models, evaluate monosemanticity using LLM-as-a-Judge with manual verification (see Appendix C.7 for details and case studies). As shown in Fig. 6, atoms with high faithfulness and stability consistently exhibit stronger monosemanticity.

These findings reveal that LLMs contain FRUs and offer a new perspective on their internal representations.

## 5. Related Work

**Neurons.** Early interpretability studies regarded neurons, the minimal computational units of neural networks, as the basic units of analysis. Subsequent work sought to ascribe functional interpretations to individual neurons (Bills et al., 2023; Geva et al., 2020). However, neurons face the polysemantic problem, activating for multiple semantically unrelated patterns (Olah et al., 2020), a phenomenon attributed to superposition (Elhage et al., 2022). These findings indicate that neurons are unsuited as such units, motivating a shift from neurons to features (Olah et al., 2020).

**Features.** Although features initially lacked a unified formal definition (Elhage et al., 2022), they are commonly understood as linear directions with specific meaning (Hewitt & Manning, 2019; Park et al., 2023; Gurnee et al., 2023; Chen et al., 2025). Sparse autoencoders (SAEs) were introduced to learn such features (Cunningham et al., 2023) and subsequently scaled to larger settings (Gao et al., 2024; Templeton, 2024). Rajamanoharan et al. (2024a) and Rajamanoharan et al. (2024b) optimized the architecture to

mitigating feature shrinkage (Wright & Sharkey, 2024). Despite widespread adoption (Lieberum et al., 2024; He et al., 2024), existing SAEs remain limited by incomplete reconstruction, with the unreconstructed component termed "dark matter" (Engels et al., 2024), and instability from feature splitting and merging (Bussmann et al., 2025; Chanin et al., 2025), undermining their suitability as FRUs. We therefore propose atoms as FRUs and develop Atom Theory.

**Other related perspectives.** Our work focuses on neurons and features as two major interpretability paradigms, comparing and unifying them under the criterion of FRUs. Other interpretability studies consider higher-level objects such as attention heads (Olsson et al., 2022; Geva et al., 2023), circuits (Meng et al., 2022; Yao et al., 2024), and localized model components (Chen et al., 2025; Hu et al., 2024). Moreover, broader work has studied AI systems from the perspective of compressed sensing (Feng et al., 2014; Giryes et al., 2016; Papyan et al., 2017; Bank & Giryes, 2018; Liang & Xiao, 2020; Wang et al., 2022; Li et al., 2023), offering a related perspective distinct from our focus on representational units.

## 6. Discussion

**Applicability of Atom Theory**   Although the empirical analysis of Atom Theory in this paper primarily focuses on LLMs, the theory itself is not inherently restricted to LLMs and may be extended to other types of neural networks, such as diffusion models. Specifically, the theory is grounded in representational distinguishability. We regard distinguishability as a first principle for analyzing neural representations: if representations encoding different information cannot be distinguished, then the collapsed representations cannot effectively carry or express the corresponding information. Moreover, further analysis of Eq. 3.1 suggests that the motivation for introducing the AIP lies in the gauge symmetry of representation space induced by matrix multiplication, where different coordinate parameterizations can represent the same function. This implies that the coordinate parameterization of representations is not unique, and the AIP is introduced to characterize this representational geometry. Since matrix multiplication is fundamental and ubiquitous in neural networks, such gauge symmetry can be expected to arise broadly across diverse neural architectures. This is consistent with our empirical observations on language models, as well as preliminary observations on diffusion models that are not included as formal results in this paper. Therefore, the AIP is not merely applicable to language models, but can serve as a more general tool for analyzing the representational geometry of neural networks. In summary, if neural representations encoding different information are expected to be distinguishable, Atom Theory has broader potential applicability beyond language models.

**Training Dynamics and Model Selection**   Theoretically, ideal atoms correspond to representational structures that achieve both high faithfulness and stability. In practice, due to limitations in optimization and data, this ideal state cannot be attained exactly, but can be approached. Consequently, candidate solutions on the Pareto frontier are typically close to the ideal regime. In this regime, the differences among candidates are usually small, and the resulting representational structures are functionally similar. Thus, model selection is not arbitrary or subjective, but rather resembles choosing from an approximate equivalence class of solutions that satisfy the theoretical criteria. This differs substantially from sparse autoencoders trained on activations from large-scale corpora. The Pareto frontiers obtained by such sparse autoencoders are often farther from the ideal regime and typically exhibit a stronger trade-off. We conjecture that this phenomenon is related to the training objective: fitting large-scale corpus activations with a finite-capacity sparse autoencoder often allows the model to capture only the most dominant subset of representational structures. Accordingly, the theoretical characterization of this training paradigm remains an open question.

**Extended Analysis and Applications**   The primary goal of this paper is to establish a theoretical framework for characterizing, evaluating, and identifying FRUs of LLMs. Accordingly, we focus on three aspects: (i) definitions and theoretical characterization; (ii) the development of evaluation metrics; and (iii) systematic validation across models, rather than optimization for specific downstream applications. Prior work has shown that representation analysis can support interpretability, intervention, behavior localization, and model safety. However, existing analyses often fall short of systematically characterizing how internal representations support complex behaviors in a structured manner. In contrast, FRUs provide a systematic view for understanding neural representations from the perspective of functional organization. Future work may construct large-scale representation repositories to systematically study the mechanisms underlying edge cases, anomalous behaviors, and safety-sensitive scenarios. From this perspective, developing scalable methods to incrementally construct such repositories remains an important direction.

## 7. Conclusion

This paper introduces and validates Atom Theory for characterizing the fundamental representational units (FRUs) of large language models (LLMs). We formalize atoms, establish criteria for ideal atoms, and prove their identifiability via threshold-activated sparse autoencoders (TSAEs). Empirically, we show that LLMs widely contain FRUs with high faithfulness and stability, exhibiting strong monosemanticity. Overall, these results elevate interpretability from heuristic analysis to a principled theory of FRUs in LLMs.

## Acknowledgements

This work was supported by the National Natural Science Foundation of China (No.U24A20335, No.62406321), the independent research project of the Key Laboratory of Cognition and Decision Intelligence for Complex Systems, and CIPS-SMP-Zhipu Large Model Fund.

## Impact Statement

This work contributes to the interpretability of large language models by introducing Atom Theory to identify fundamental representational units. Improved understanding of internal representations may enhance transparency, safety, and reliability of AI systems, supporting model auditing and alignment. We expect the net impact to be positive.

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

# A. Proofs

## A.1. Explanation for Equation 3.1

For

$$W_U' \leftarrow A^{-\top} W_U + \boldsymbol{b} \boldsymbol{1}^\top, \; \boldsymbol{h}'^L \leftarrow A\boldsymbol{h}^L, \tag{3.1}$$

we provide a simple derivation as follows:

$$W_U'^\top \boldsymbol{h}' = (A^{-\top} W_U + \boldsymbol{b} \boldsymbol{1}^\top)^\top (A\boldsymbol{h}) \tag{A.1}$$

$$= W_U^\top (A^{-1} A)\boldsymbol{h} + \boldsymbol{1}(\boldsymbol{b}^\top A\boldsymbol{h}) \tag{A.2}$$

$$= W_U^\top \boldsymbol{h} + c(\boldsymbol{h})\boldsymbol{1}, \tag{A.3}$$

where $c(\boldsymbol{h}) = \boldsymbol{b}^\top A\boldsymbol{h} \in \mathbb{R}$ is a scalar. Using the translation invariance property of softmax, $\text{Softmax}(\boldsymbol{z}+c\boldsymbol{1}) = \text{Softmax}(\boldsymbol{z})$, the result follows.

It is important to note that this is the only form, meaning that $W_U$ can only be identified up to an invertible transformation plus a bias, and $\boldsymbol{h}$ can only be identified up to an invertible transformation.

## A.2. Proof of Theorem 3.2

**Theorem 3.2** (Explicit Form of the Atomic Inner Product). *Let $\langle \boldsymbol{d}_i, \boldsymbol{d}_j \rangle_S = \boldsymbol{d}_i^\top \boldsymbol{S} \boldsymbol{d}_j$ be an atomic inner product with $\boldsymbol{S} \in \mathbb{R}^{H \times H}$ symmetric and positive definite. If the columns of $\boldsymbol{D} = [\boldsymbol{d}_1, \boldsymbol{d}_2, \cdots, \boldsymbol{d}_{|D|}]$ form the atom set such that $\forall i, \|\boldsymbol{d}_i\|_S = c > 0$, and $\mathcal{D} \simeq \mathbb{R}^H$, then $\boldsymbol{S} = c^2 (\boldsymbol{D}\boldsymbol{D}^\top)^{-1}$.*

*Proof.* Since $\langle \cdot, \cdot \rangle_S$ is atomic inner product, for any pair of atoms $\boldsymbol{d}_i$ and $\boldsymbol{d}_j$ we have

$$\boldsymbol{d}_i^\top \boldsymbol{S} \boldsymbol{d}_j = \langle \boldsymbol{d}_i, \boldsymbol{d}_j \rangle_S = \begin{cases} 0 & i \neq j, \\ c^2 & i = j. \end{cases} \tag{A.4}$$

Applying this property to the atom set $D = [\boldsymbol{d}_1, \boldsymbol{d}_2, \cdots, \boldsymbol{d}_{|D|}]$ yields

$$c^2 I = D^\top S D. \tag{A.5}$$

Since $\{\boldsymbol{d}_1, \boldsymbol{d}_2, \cdots, \boldsymbol{d}_{|D|}\}$ spans the atomic space $\mathcal{D}$, and $\mathcal{D} \simeq \mathbb{R}^H$, it follows that $\text{rank}(D) = H$. Let $S^{1/2}$ denote the symmetric positive-definite square root of $S$. Then

$$D^\top S D = (S^{1/2} D)^\top (S^{1/2} D), \tag{A.6}$$

which implies $\text{rank}(I) = \text{rank}(D^\top S D) = \text{rank}(S^{1/2} D) = \text{rank}(D)$. Therefore, $\text{rank}(D) = |D| \leq H$. Combining this with the earlier condition gives $|D| = \text{rank}(D) = H$, which shows that $D$ is invertible. Consequently,

$$S = c^2 (DD^\top)^{-1}. \tag{A.7}$$

$\square$

## A.3. Proof of Corollary 3.3

**Corollary 3.3** (Normalized Atomic Inner Product; NAIP). *Let the atomic inner product be defined by $\langle \boldsymbol{d}_i, \boldsymbol{d}_j \rangle_S = \boldsymbol{d}_i^\top \boldsymbol{S} \boldsymbol{d}_j$, where $\boldsymbol{S}$ is symmetric and positive definite. Suppose the columns of $\boldsymbol{D} = [\boldsymbol{d}_1, \cdots, \boldsymbol{d}_{|D|}]$ form the atom set satisfying $\forall i, \|\boldsymbol{d}_i\|_S = c > 0$. Then, for any $i, j$,*

$$\langle \boldsymbol{d}_i, \boldsymbol{d}_j \rangle_{\tilde{S}} := \frac{\langle \boldsymbol{d}_i, \boldsymbol{d}_j \rangle_S}{\|\boldsymbol{d}_i\|_S \|\boldsymbol{d}_j\|_S} = \boldsymbol{d}_i^\top \tilde{\boldsymbol{S}} \boldsymbol{d}_j, \quad \tilde{\boldsymbol{S}} = (\boldsymbol{D}\boldsymbol{D}^\top)^{-1}. \tag{3.2}$$

*Consequently, the bilinear form $\langle \boldsymbol{d}_i, \boldsymbol{d}_j \rangle_{\tilde{S}} = \boldsymbol{d}_i^\top \tilde{\boldsymbol{S}} \boldsymbol{d}_j$ defines the **normalized atomic inner product**.*

*Proof.* Since $\langle \boldsymbol{d}_i, \boldsymbol{d}_j \rangle_S = 0$ for $i \neq j$, and $\|\boldsymbol{d}_i\|_S = \|\boldsymbol{d}_j\|_S = c > 0$, it follows that $D^\top S D = c^2 I$. Therefore, $\tilde{S} = \frac{1}{c^2} S$ satisfies $D^\top \tilde{S} D = I$, which implies that the atoms are orthonormal. Since $D$ is invertible, we also have $\tilde{S} = D^{-\top} D^{-1} = (DD^\top)^{-1}$, and thus $\langle \cdot, \cdot \rangle_{\tilde{S}}$ is a symmetric positive-definite inner product. $\square$

## A.4. Proof of Theorem 3.8

**Theorem 3.8** (Coherence–RIP Upper Bound). *Let $\tilde{D} \in \mathbb{R}^{H \times |D|}$ and define the coherence $\mu := \max_{i \neq j} |\langle \tilde{d}_i, \tilde{d}_j \rangle| \leq \varepsilon$. For any $K$-sparse vector $\delta \in \mathbb{R}^{|D|}$ with $\|\delta\|_0 \leq K$,*

$$\left(1 - (K-1)\mu\right) \|\delta\|_2^2 \leq \|\tilde{D}\delta\|_2^2 \leq \left(1 + (K-1)\mu\right) \|\delta\|_2^2. \tag{3.5}$$

*Hence $\delta_K(\tilde{D}) \leq (K-1)\mu$; in particular, $\tilde{D}$ satisfies the $K$-RIP whenever $(K-1)\mu < 1$.*

*Proof.* Let $\text{supp}(\delta) = \mathcal{S} \subseteq [|D|]$ and $|\mathcal{S}| \leq K$. Then, we have

$$\|\tilde{D}\delta\|_2^2 = \delta^\top (\tilde{D}^\top \tilde{D}) \delta = \sum_{i \in \mathcal{S}} \delta_i^2 + 2 \sum_{\substack{i < j \\ i,j \in \mathcal{S}}} \delta_i \delta_j \langle \tilde{d}_i, \tilde{d}_j \rangle. \tag{A.8}$$

By the fact that $|\langle \tilde{d}_i, \tilde{d}_j \rangle| \leq \mu$ and applying the triangle inequality, we obtain:

$$\|\tilde{D}\delta\|_2^2 \geq \sum_{i \in \mathcal{S}} \delta_i^2 - 2\mu \sum_{i < j} |\delta_i \delta_j|, \tag{A.9}$$

$$\|\tilde{D}\delta\|_2^2 \leq \sum_{i \in \mathcal{S}} \delta_i^2 + 2\mu \sum_{i < j} |\delta_i \delta_j|. \tag{A.10}$$

Next, we observe that:

$$\left( \sum_{i \in \mathcal{S}} |\delta_i| \right)^2 = \sum_{i \in \mathcal{S}} \delta_i^2 + 2 \sum_{i < j} |\delta_i \delta_j| \leq |\mathcal{S}| \sum_{i \in \mathcal{S}} \delta_i^2 \leq K \sum_{i \in \mathcal{S}} \delta_i^2. \tag{A.11}$$

Thus, we have $2 \sum_{i < j} |\delta_i \delta_j| \leq (K-1) \sum_{i \in \mathcal{S}} \delta_i^2$. Substituting this back, we conclude the proof. $\square$

## A.5. Proof of Theorem 3.9

**Theorem 3.9** (Uniqueness and Exact $\ell_1$ Recoverability). *Let $\tilde{D} \in \mathbb{R}^{H \times |D|}$ and define the coherence $\mu := \max_{i \neq j} |\langle \tilde{d}_i, \tilde{d}_j \rangle| \leq \varepsilon$. If $\mu < \frac{1}{2K-1}$, then for every $\delta \in \mathbb{R}^{|D|}$ with $\|\delta\|_0 \leq K$, the $K$-sparse representation determined by $\tilde{m} = \tilde{D}\delta$ is unique; that is, no other $K$-sparse vector yields the same $\tilde{m}$. Moreover, $\delta$ is the unique minimizer of the convex program*

$$\min_{x \in \mathbb{R}^{|D|}} \|x\|_1 \quad \text{subject to} \quad \tilde{D}x = \tilde{m}. \tag{3.6}$$

*Proof.* We first prove that under the condition $\mu < \frac{1}{2K-1}$, the $K$-sparse representation is unique.

Suppose there exist two distinct $K$-sparse coefficient vectors $\delta, \delta'$ such that $\tilde{D}\delta = \tilde{D}\delta'$. Let $h = \delta - \delta' \neq 0$. Then $\tilde{D}h = 0$ and $\|h\|_0 \leq 2K$. By Theorem 3.8 (applied with $K$ replaced by $2K$), we have

$$\left(1 - (2K-1)\mu\right) \|h\|_2^2 \leq \|\tilde{D}h\|_2^2 = 0. \tag{A.12}$$

If $\mu < \frac{1}{2K-1}$, then the prefactor on the left is strictly positive, which forces $\|h\|_2 = 0$. This contradicts $h \neq 0$. Hence uniqueness holds.

Next, we prove that under the same condition $\mu < \frac{1}{2K-1}$, the sparse vector $\delta$ is also the unique solution of the convex optimization problem

$$\min_{x \in \mathbb{R}^{|D|}} \|x\|_1 \quad \text{s.t.} \quad \tilde{D}x = \tilde{m}. \tag{A.13}$$

The overall strategy is as follows: (i) show that the null space property of order $K$ ($\text{NSP}_K$) holds under the assumption $\mu < \frac{1}{2K-1}$; (ii) recall the equivalence $\text{NSP}_K \iff$ exact and unique recovery of any $K$-sparse solution via noiseless $\ell_1$-minimization.

Formally, the null space property of order $K$ ($\text{NSP}_K$) is defined as

$$\forall h \in \ker(\tilde{D}) \setminus \{0\}, \ \forall \mathcal{S} \subseteq [n], \ |\mathcal{S}| \leq K : \quad \boxed{\|h_{\mathcal{S}}\|_1 < \|h_{\mathcal{S}^c}\|_1}, \tag{A.14}$$

where $\ker(\tilde{D}) = \{h : \tilde{D}h = 0\}$, $\mathcal{S} \subseteq [n]$ is an index set with $[n] = \{1, \ldots, n\}$, $h_{\mathcal{S}}$ denotes the restriction of $h$ to the coordinates in $\mathcal{S}$ (with other entries set to zero), and $\mathcal{S}^c = [n] \setminus \mathcal{S}$ is a complementary set.

**Step (i): Proof of NSP$_K$.**  Let $G = \tilde{D}^\top \tilde{D}$ denote the Gram matrix. Since each column of $\tilde{D}$ is normalized, we have $G_{jj} = 1$ and $|G_{ij}| \leq \mu$ for $i \neq j$. Take any $\boldsymbol{h} \in \ker(\tilde{D}) \setminus \{\boldsymbol{0}\}$ and any index set $\mathcal{S}$ with $|\mathcal{S}| = K$.

Since $\tilde{D}\boldsymbol{h} = 0$, we have $G\boldsymbol{h} = 0$. For any $j$,

$$0 = (G\boldsymbol{h})_j = \sum_i G_{ji}h_i = G_{jj}h_j + \sum_{i \neq j} G_{ji}h_i \quad \Rightarrow \quad h_j = -\sum_{i \neq j} G_{ji}h_i. \tag{A.15}$$

Taking absolute values and using $|G_{ji}| \leq \mu$, we obtain

$$|h_j| \leq \mu \sum_{i \neq j} |h_i|. \tag{A.16}$$

Summing over $j \in \mathcal{S}$ gives

$$\sum_{j \in \mathcal{S}} |h_j| \ \leq \ \mu \sum_{j \in \mathcal{S}} \sum_{i \neq j} |h_i|. \tag{A.17}$$

The inner summation can be decomposed into contributions from $i \in \mathcal{S} \setminus \{j\}$ and $i \in \mathcal{S}^c$:

$$\sum_{j \in \mathcal{S}} \sum_{i \neq j} |h_i| = \underbrace{\sum_{j \in \mathcal{S}} \sum_{\substack{i \in \mathcal{S} \\ i \neq j}} |h_i|}_{\text{each } i \in \mathcal{S} \text{ counted } K-1 \text{ times}} + \underbrace{\sum_{j \in \mathcal{S}} \sum_{i \in \mathcal{S}^c} |h_i|}_{\text{each } i \in \mathcal{S}^c \text{ counted } K \text{ times}} . \tag{A.18}$$

Hence,

$$\|\boldsymbol{h}_{\mathcal{S}}\|_1 \ \leq \ \mu\big((K-1)\|\boldsymbol{h}_{\mathcal{S}}\|_1 + K\|\boldsymbol{h}_{\mathcal{S}^c}\|_1\big). \tag{A.19}$$

Rearranging,

$$\big(1 - (K-1)\mu\big)\|\boldsymbol{h}_{\mathcal{S}}\|_1 \ \leq \ K\mu\,\|\boldsymbol{h}_{\mathcal{S}^c}\|_1. \tag{A.20}$$

Dividing through by the positive factor $1 - (K-1)\mu$, define

$$\alpha := \frac{K\mu}{1 - (K-1)\mu}. \tag{A.21}$$

When $\mu < \frac{1}{2K-1}$, we have $\alpha < 1$. Since $\boldsymbol{h} \neq 0$ and $\tilde{D}\boldsymbol{h} = 0$, it is impossible for $\|\boldsymbol{h}_{\mathcal{S}^c}\|_1 = 0$ (otherwise both terms would vanish, forcing $\boldsymbol{h} = 0$, a contradiction). Therefore,

$$\|\boldsymbol{h}_{\mathcal{S}}\|_1 \ \leq \ \alpha\|\boldsymbol{h}_{\mathcal{S}^c}\|_1 \ < \ \|\boldsymbol{h}_{\mathcal{S}^c}\|_1. \tag{A.22}$$

Take any $\mathcal{S}_0$ with $|\mathcal{S}_0| = k \leq K$, and extend it to a superset $\mathcal{S} \supseteq \mathcal{S}_0$ such that $|\mathcal{S}| = K$. Then,

$$\|\boldsymbol{h}_{\mathcal{S}_0}\|_1 \ \leq \ \|\boldsymbol{h}_{\mathcal{S}}\|_1, \qquad \|\boldsymbol{h}_{\mathcal{S}_0^c}\|_1 \ \geq \ \|\boldsymbol{h}_{\mathcal{S}^c}\|_1. \tag{A.23}$$

If we know that $\|\boldsymbol{h}_{\mathcal{S}}\|_1 < \|\boldsymbol{h}_{\mathcal{S}^c}\|_1$ holds for all $\mathcal{S}$ of size $K$, then it follows that

$$\|\boldsymbol{h}_{\mathcal{S}_0}\|_1 \ \leq \ \|\boldsymbol{h}_{\mathcal{S}}\|_1 \ < \ \|\boldsymbol{h}_{\mathcal{S}^c}\|_1 \ \leq \ \|\boldsymbol{h}_{\mathcal{S}_0^c}\|_1. \tag{A.24}$$

Thus, the inequality also holds for any $\mathcal{S}_0$ with $|\mathcal{S}_0| \leq K$, which establishes NSP$_K$.

**Step (ii): Equivalence between NSP$_K$ and $\ell_1$ recovery.**

**NSP$_K$ $\Rightarrow$ unique $\ell_1$ recovery:**  Suppose $\hat{\boldsymbol{x}}$ is another feasible solution such that $\tilde{D}\hat{\boldsymbol{x}} = \tilde{D}\boldsymbol{\delta}$. Let $\boldsymbol{h} = \hat{\boldsymbol{x}} - \boldsymbol{\delta} \in \ker(\tilde{D}) \setminus \{\boldsymbol{0}\}$, and let $\mathcal{S} = \operatorname{supp}(\boldsymbol{\delta})$ with $|\mathcal{S}| \leq K$. Then

$$\|\hat{\boldsymbol{x}}\|_1 = \|\boldsymbol{\delta} + \boldsymbol{h}\|_1 = \|\boldsymbol{\delta}_{\mathcal{S}} + \boldsymbol{h}_{\mathcal{S}}\|_1 + \|\boldsymbol{h}_{\mathcal{S}^c}\|_1 \ \geq \ \|\boldsymbol{\delta}_{\mathcal{S}}\|_1 - \|\boldsymbol{h}_{\mathcal{S}}\|_1 + \|\boldsymbol{h}_{\mathcal{S}^c}\|_1 \ > \ \|\boldsymbol{\delta}_{\mathcal{S}}\|_1 = \|\boldsymbol{\delta}\|_1, \tag{A.25}$$

where the strict inequality follows from NSP$_K$. Hence, $\boldsymbol{\delta}$ is the unique minimizer of the $\ell_1$ problem.

**Unique $\ell_1$ recovery $\Rightarrow$ NSP$_K$:** We argue by contradiction. If NSP$_K$ does not hold, then there exists $\boldsymbol{h} \in \ker(\tilde{D}) \setminus \{\boldsymbol{0}\}$ and some $\mathcal{S}$ with $|\mathcal{S}| \leq K$ such that $\|\boldsymbol{h}_{\mathcal{S}}\|_1 \geq \|\boldsymbol{h}_{\mathcal{S}^c}\|_1$. Take any nonzero $K$-sparse $\boldsymbol{\delta}$ with $\text{supp}(\boldsymbol{\delta}) = \mathcal{S}$, and choose $\delta_j = \alpha_j \, \text{sgn}(h_j)$ with $\alpha_j \geq |h_j|$ coordinate-wise. Consider $\hat{\boldsymbol{x}} = \boldsymbol{\delta} - \boldsymbol{h}$. Since $\tilde{D}\boldsymbol{h} = \boldsymbol{0}$, both $\boldsymbol{\delta}$ and $\hat{\boldsymbol{x}}$ are feasible, and

$$\|\hat{\boldsymbol{x}}\|_1 = \|\boldsymbol{\delta}_{\mathcal{S}} - \boldsymbol{h}_{\mathcal{S}}\|_1 + \|\boldsymbol{h}_{\mathcal{S}^c}\|_1 = \|\boldsymbol{\delta}_{\mathcal{S}}\|_1 - \|\boldsymbol{h}_{\mathcal{S}}\|_1 + \|\boldsymbol{h}_{\mathcal{S}^c}\|_1 \; \leq \; \|\boldsymbol{\delta}_{\mathcal{S}}\|_1 = \|\boldsymbol{\delta}\|_1. \tag{A.26}$$

Thus, $\boldsymbol{\delta}$ is not the unique minimizer of the $\ell_1$ problem (and may even fail to be a minimizer). This contradicts the uniqueness assumption. Therefore, NSP$_K$ must hold. $\qquad\square$

### A.6. Proof of Corollary 3.10

**Corollary 3.10** (Monorepresentationality / Injectivity). *Under the condition $\mu < \frac{1}{2K-1}$ of Theorem 3.9, define $\Sigma_K := \{\boldsymbol{\delta} \in \mathbb{R}^{|D|} : \|\boldsymbol{\delta}\|_0 \leq K\}$ and $\Phi : \Sigma_K \to \mathbb{R}^H$ by $\Phi(\boldsymbol{\delta}) := \tilde{D}\boldsymbol{\delta}$. Then $\Phi$ is injective on $\Sigma_K$. That is, for any $\boldsymbol{x}, \boldsymbol{y} \in \Sigma_K$, if $\tilde{D}\boldsymbol{x} = \tilde{D}\boldsymbol{y}$, it follows that $\boldsymbol{x} = \boldsymbol{y}$.*

*Proof.* Take arbitrary $\boldsymbol{x}, \boldsymbol{y} \in \Sigma_K$ such that $\tilde{D}\boldsymbol{x} = \tilde{D}\boldsymbol{y}$. Let $\tilde{\boldsymbol{m}} := \tilde{D}\boldsymbol{x}$, then clearly $\tilde{\boldsymbol{m}} = \tilde{D}\boldsymbol{y}$, with $\|\boldsymbol{x}\|_0 \leq K$ and $\|\boldsymbol{y}\|_0 \leq K$.

By Theorem 3.9, under the condition $\mu < \frac{1}{2K-1}$, the $K$-sparse representation $\boldsymbol{\delta}$ determined by the equation $\tilde{\boldsymbol{m}} = \tilde{D}\boldsymbol{\delta}$ is unique: among all coefficient vectors satisfying $\|\boldsymbol{\delta}\|_0 \leq K$, there exists only one that generates $\tilde{\boldsymbol{m}}$.

Since both $\boldsymbol{x}$ and $\boldsymbol{y}$ are feasible solutions satisfying $\|\boldsymbol{\delta}\|_0 \leq K$ and $\tilde{D}\boldsymbol{\delta} = \tilde{\boldsymbol{m}}$, uniqueness forces $\boldsymbol{x} = \boldsymbol{y}$. Hence, $\Phi$ is injective on $\Sigma_K$. $\qquad\square$

### A.7. Proof of Theorem 3.11

**Theorem 3.11** (Identifiability of Threshold-activated SAEs; TSAEs). *Let $M = \{\boldsymbol{m}_i\}_{i=1}^{|M|} \subset \mathbb{R}^H$ with $\boldsymbol{m}_i = D\boldsymbol{\delta}_i$, where $D = [\boldsymbol{d}_1, \cdots, \boldsymbol{d}_{|D|}] \in \mathbb{R}^{H \times |D|}$ satisfies $|\langle \tilde{\boldsymbol{d}}_i, \tilde{\boldsymbol{d}}_j \rangle| \leq \epsilon$ for all $i \neq j$. Suppose each $\boldsymbol{\delta}_i \in \mathbb{R}^{|D|}$ is $K$-sparse, i.e. $\|\boldsymbol{\delta}_i\|_0 \leq K$. Consider the threshold activation function*

$$\sigma_\tau(x) = \begin{cases} 0 & x < \tau, \\ x & x \geq \tau, \end{cases} \tag{3.7}$$

*with threshold $\tau > 0$. Assume there exist constants $0 < \delta_{\min} \leq \delta_{\max}$ such that, for each support $\mathcal{S}_i = \text{supp}(\boldsymbol{\delta}_i)$, $\delta_{\min} \leq \delta_{ij} \leq \delta_{\max}$, $\forall j \in \mathcal{S}_i$. If the amplitude gap and threshold satisfy $\varepsilon K \delta_{\max} < \tau < \delta_{\min} - \varepsilon(K-1)\delta_{\max}$, which is feasible whenever $\delta_{\min} > \varepsilon(2K-1)\delta_{\max}$, then setting TSAEs with $\boldsymbol{W}_{dec} = D$ and $\boldsymbol{W}_{enc} = D^\top \tilde{\boldsymbol{S}}$ yields*

$$\forall i, \quad \boldsymbol{W}_{dec}\, \sigma_\tau(\boldsymbol{W}_{enc}\boldsymbol{m}_i) = \boldsymbol{m}_i. \tag{3.8}$$

*This parameterization exactly recover the atom set $D$.*

*Proof.* Consider a single-layer linear–nonlinear encoder of the form $W_{\text{dec}}\, \sigma_\tau(W_{\text{enc}}\boldsymbol{m}_i)$, with training objective

$$W_{\text{dec}}\, \sigma_\tau(W_{\text{enc}}\boldsymbol{m}_i) = \boldsymbol{m}_i, \quad \forall i. \tag{A.27}$$

Set $W_{\text{dec}} = D$ and $W_{\text{enc}} = D^\top \tilde{S}$. Denote $\mathcal{S}_i = \text{supp}(\boldsymbol{\delta}_i)$. Then

$$D^\top \tilde{S}\boldsymbol{m}_i = \begin{bmatrix} \boldsymbol{d}_1^\top \\ \boldsymbol{d}_2^\top \\ \vdots \\ \boldsymbol{d}_{|D|}^\top \end{bmatrix} \tilde{S} \begin{bmatrix} \boldsymbol{d}_1 & \boldsymbol{d}_2 & \cdots & \boldsymbol{d}_{|D|} \end{bmatrix} \boldsymbol{\delta}_i \tag{A.28}$$

$$= \begin{bmatrix} \boldsymbol{d}_1^\top \tilde{S}\boldsymbol{d}_1 & \cdots & \boldsymbol{d}_1^\top \tilde{S}\boldsymbol{d}_{|D|} \\ \boldsymbol{d}_2^\top \tilde{S}\boldsymbol{d}_1 & \cdots & \boldsymbol{d}_2^\top \tilde{S}\boldsymbol{d}_{|D|} \\ \vdots & \ddots & \vdots \\ \boldsymbol{d}_{|D|}^\top \tilde{S}\boldsymbol{d}_1 & \cdots & \boldsymbol{d}_{|D|}^\top \tilde{S}\boldsymbol{d}_{|D|} \end{bmatrix} \boldsymbol{\delta}_i \tag{A.29}$$

$$= G\,\boldsymbol{\delta}_i, \quad G := D^\top \tilde{S}D. \tag{A.30}$$

By NAIP, we have $G_{kk} = 1$ and for $k \neq j$, $|G_{kj}| = |\langle \tilde{\boldsymbol{d}}_k, \tilde{\boldsymbol{d}}_j \rangle| \leq \varepsilon$. Thus, for any index $k$,

$$(G\boldsymbol{\delta}_i)_k = \begin{cases} \delta_{ik} + \underbrace{\sum_{j \in \mathcal{S}_i \backslash \{k\}} \delta_{ij} G_{kj}}_{=:e_{ik}}, & k \in \mathcal{S}_i, \\ \underbrace{\sum_{j \in \mathcal{S}_i} \delta_{ij} G_{kj}}_{=:e_{ik}}, & k \notin \mathcal{S}_i. \end{cases} \tag{A.31}$$

Using the coherence bound, we obtain the deterministic perturbation estimate

$$\begin{cases} (G\boldsymbol{\delta}_i)_k \geq \delta_{ik} - \varepsilon(K-1)\delta_{\max}, & k \in \mathcal{S}_i, \\ (G\boldsymbol{\delta}_i)_k \leq \varepsilon K \delta_{\max}, & k \notin \mathcal{S}_i. \end{cases} \tag{A.32}$$

Choose a threshold $\tau$ such that

$$\varepsilon K \delta_{\max} < \tau < \delta_{\min} - \varepsilon(K-1)\delta_{\max}. \tag{A.33}$$

This ensures support separation

$$\begin{cases} (G\boldsymbol{\delta}_i)_k > \tau, & k \in \mathcal{S}_i, \\ (G\boldsymbol{\delta}_i)_k < \tau, & k \notin \mathcal{S}_i. \end{cases} \tag{A.34}$$

Therefore, the coordinate-wise nonlinearity

$$\sigma_\tau(x) := \begin{cases} 0 & x < \tau, \\ x & x \geq \tau \end{cases} \tag{A.35}$$

produces activations $\boldsymbol{z}_i := \sigma_\tau(G\boldsymbol{\delta}_i)$, with $\operatorname{supp}(\boldsymbol{z}_i) = \mathcal{S}_i$. For $k \in \mathcal{S}_i$,

$$z_{ik} = (G\boldsymbol{\delta}_i)_k = \delta_{ik} + e_{ik}. \tag{A.36}$$

Since for any $j \neq k$, $G_{kj} = \boldsymbol{d}_k^\top \tilde{S} \boldsymbol{d}_j$ is distributed approximately as $\mathcal{N}(0, s^2)$ with small variance $s$, it follows that

$$\mathbb{E}[e_{ik}] = \mathbb{E}\left[ \sum_{j \in \mathcal{S}_i \backslash \{k\}} \delta_{ij} G_{kj} \right] = \sum_{j \in \mathcal{S}_i \backslash \{k\}} \delta_{ij} \, \mathbb{E}[G_{kj}] = 0. \tag{A.37}$$

Since $\mathbb{E}[e_{ik}] = 0$ for all $i, k$, the law of large numbers implies that, in probability,

$$D \, \sigma_\tau(D^\top \tilde{S} \boldsymbol{m}_i) = \boldsymbol{m}_i, \quad \forall i. \tag{A.38}$$

Thus, under this parametrization, the SAE recovers the target atom set $D$. $\qquad \square$

## B. Representation Shift

In this section, we provide further details on representation shift, including the experimental setup, ablation studies, and additional analyses, to enable a comprehensive understanding of this phenomenon.

Specifically, we randomly sample 128 subject entities from the WikiData dataset (Vrandečić & Krötzsch, 2014) and extract the corresponding activations across all layers of multiple language model families, including GPT-2 (GPT2-Small, GPT2-Medium, GPT2-Large), GPT-J (GPT-J-6B), Pythia (Pythia-1B, Pythia-1.4B, Pythia-2.8B, Pythia-6.9B), Llama-2 (Llama2-7B, Llama2-13B), Llama-3 (Llama3-8B), Llama-3.1 (Llama3.1-8B), and Gemma-2 (Gemma2-2B, Gemma2-9B), which serve as the target representations for our analysis.

For each entity, we extract activations at the position of its final token, which has been empirically identified via causal tracing as the key site of knowledge extraction in language models (Meng et al., 2022). Thus, for each layer we obtain 128

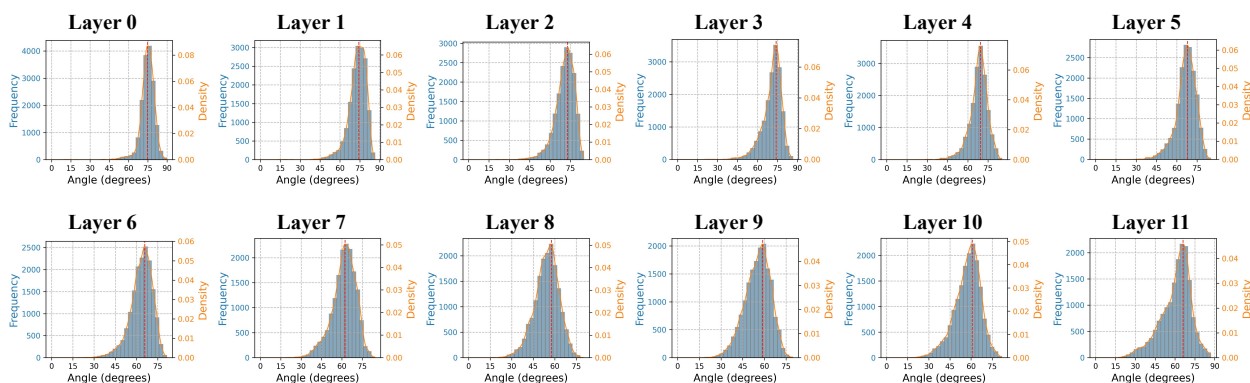

*Figure 7.* Representation shift of GPT2-Small.

representations, yielding $128 \times 128 = 16{,}384$ representation pairs for analyzing angular distributions. This sample size is sufficient to capture the overall distributional characteristics, as additional samples do not alter the distribution (Figs. 10–11), while also facilitating non-overlap with activations used in subsequent sampling.

We first compute the angles between representation pairs using the Euclidean inner product. Specifically, for representations $\boldsymbol{u}, \boldsymbol{v} \in \mathbb{R}^H$, the Euclidean cosine similarity is defined as $\cos(\boldsymbol{u}, \boldsymbol{v}) = \frac{\langle \boldsymbol{u}, \boldsymbol{v} \rangle}{\|\boldsymbol{u}\|_2 \|\boldsymbol{v}\|_2}$, which we then convert to angles for clearer visualization.

The results (Figs. 7–9) show that when angles are computed under the Euclidean inner product, the centroid of the angular distribution deviates markedly from $90°$, indicating a representation shift. Full experimental results are available in the arXiv version or upon request. This phenomenon consistently appears across all layers and model families, independent of model architectures and training corpora, and thus reflects a systematic non-uniformity in representation distributions. Such anisotropy distorts the underlying geometry of representations, which should be isotropic for random epresentations. In LLMs, this anisotropy is nearly indiscriminate: representations are globally attracted toward a dominant direction, yielding high cosine similarity even for unrelated samples. Direct evidence is provided in Figs. 10–11: increasing the sample size used to estimate the distribution leaves the centroid unchanged, confirming that representations are indeed attracted toward the same direction. Full experimental results are available in the arXiv version or upon request.

We then collect 100K activations $\boldsymbol{k}$ per layer for each model from Wikipedia (manually verified to have no overlap with the target representations) and compute $\mathbb{E}[\boldsymbol{k}\boldsymbol{k}^\top]$ to estimate the normalized atomic inner product. Under the inner-product space induced by $\tilde{S}$, the cosine similarity is defined as $\cos_{\tilde{S}}(\boldsymbol{u}, \boldsymbol{v}) = \frac{\langle \boldsymbol{u}, \boldsymbol{v} \rangle_{\tilde{S}}}{\|\boldsymbol{u}\|_{\tilde{S}} \|\boldsymbol{v}\|_{\tilde{S}}} = \frac{\boldsymbol{u}^\top \tilde{S} \boldsymbol{v}}{\sqrt{\boldsymbol{u}^\top \tilde{S} \boldsymbol{u}} \sqrt{\boldsymbol{v}^\top \tilde{S} \boldsymbol{v}}}$, where $\tilde{S}$ is estimated in practice as $(\mathbb{E}[\boldsymbol{k}\boldsymbol{k}^\top])^{-1}$. As before, cosine similarities are converted to angles for clearer visualization.

As shown in Figs. 12–14, the atomic inner product corrects the representation shift, restoring the centroid of the angular distribution to $90°$ and thereby capturing the underlying representational geometry, aligned with theoretical expectations. Full experimental results are available in the arXiv version or upon request. This effect holds consistently across all layers and model families. Further inspection shows that the long tail of the distribution (high-similarity pairs) corresponds to highly related samples, indicating that LLM representations are intrinsically isotropic when measured with the appropriate metric.

Although we identify the correct inner product for LLM representations and verify that their global geometry matches theoretical expectations, substantial superposition remains pervasive (Hu et al., 2025). For example, Fig. 15 shows that activations in Gemma2-2B still exhibit widespread superposition, indicating that these representations are not fully disentangled. The persistence of superposition suggests that raw activations are not the most suitable fundamental representational units. We therefore decompose high-dimensional representations into atoms that better satisfy the criteria for FRUs; the resulting decomposition (Fig. 16) demonstrates that the identified atoms effectively disentangle representations and resolve superposition.

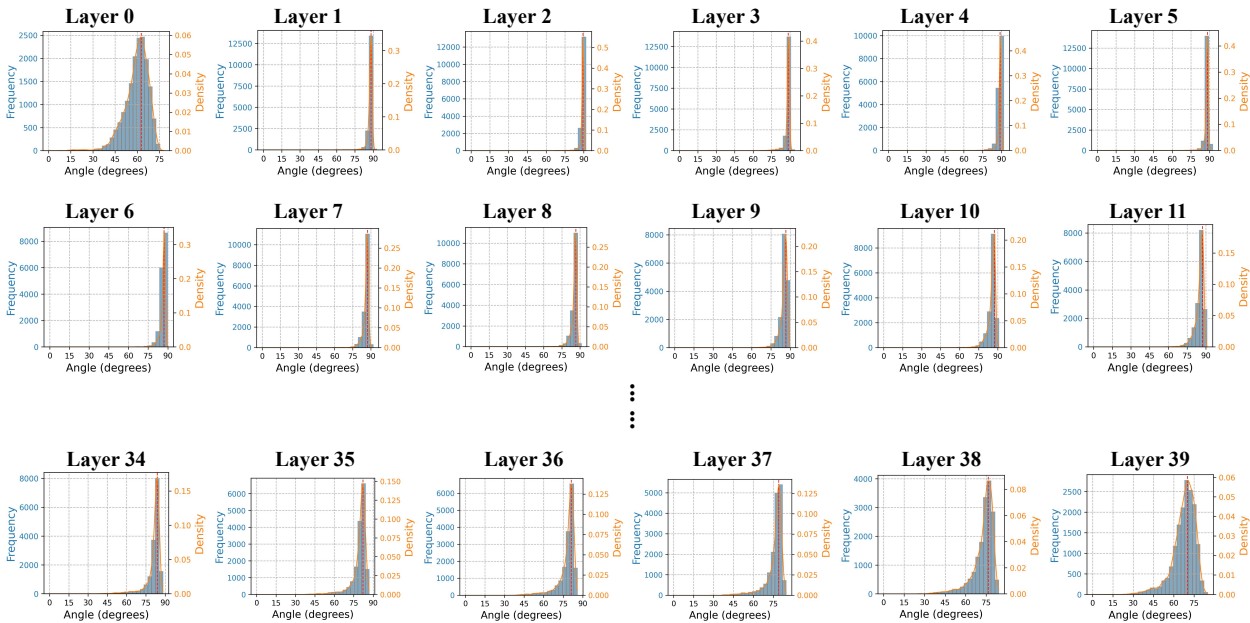

*Figure 8.* Representation shift of Llama2-13B.

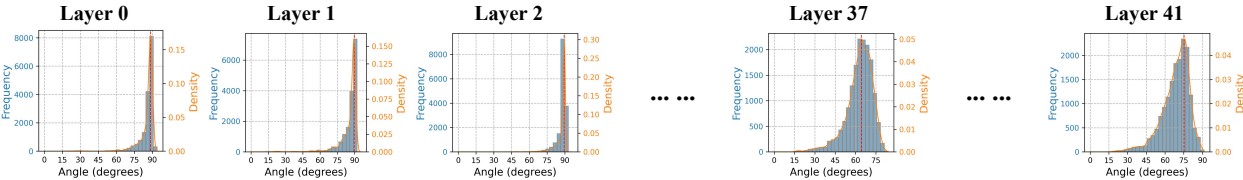

*Figure 9.* Representation shift of Gemma2-9B.

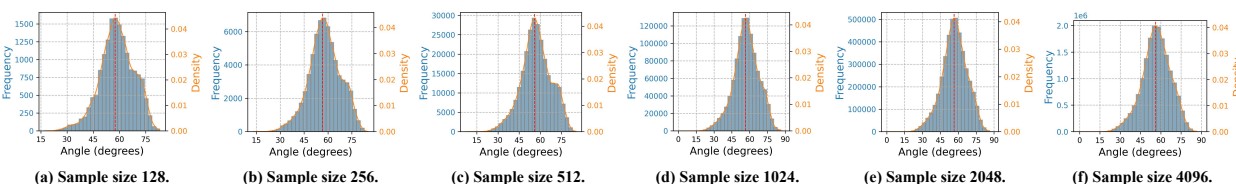

*Figure 10.* Representation shift of GPT2-Small under increasing sample sizes.

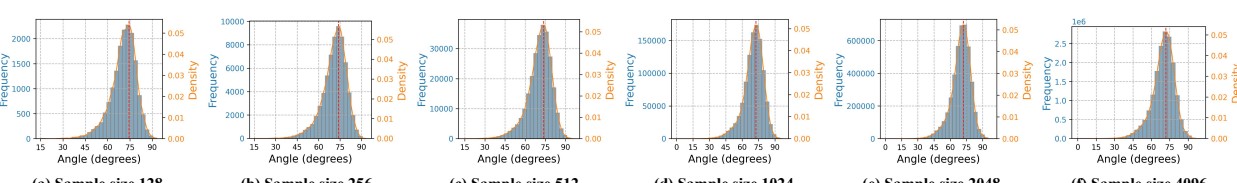

*Figure 11.* Representation shift of Gemma2-9B under increasing sample sizes.

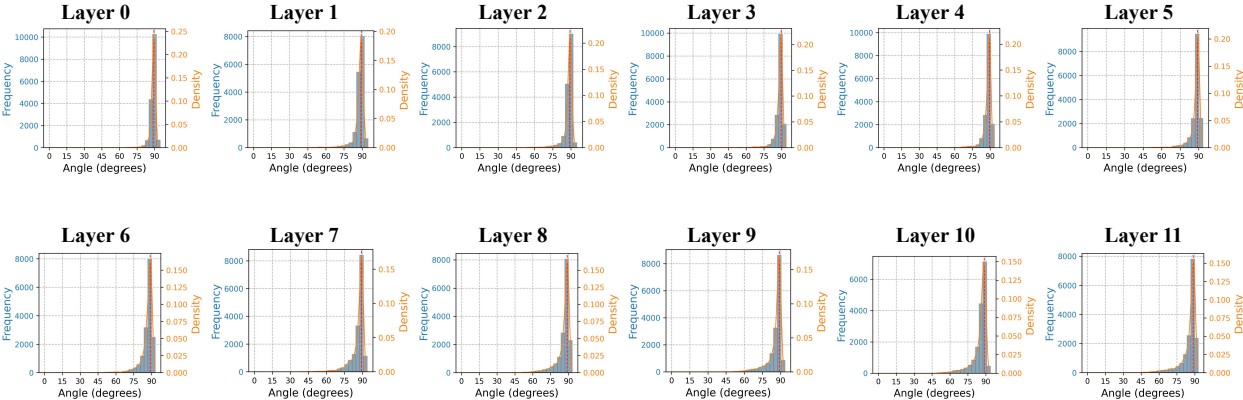

*Figure 12.* Correction of representation shift on GPT2-Small.

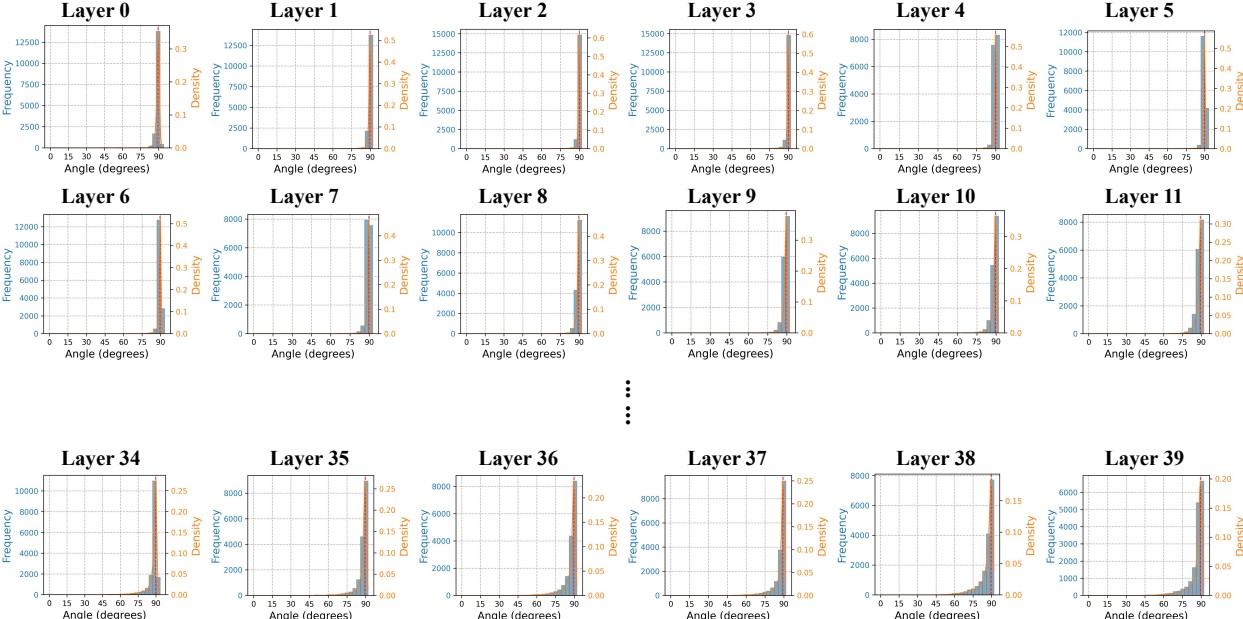

*Figure 13.* Correction of representation shift on Llama2-13B.

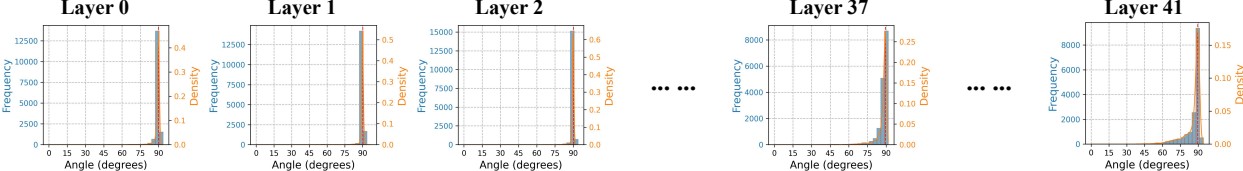

*Figure 14.* Correction of representation shift on Gemma2-9B.

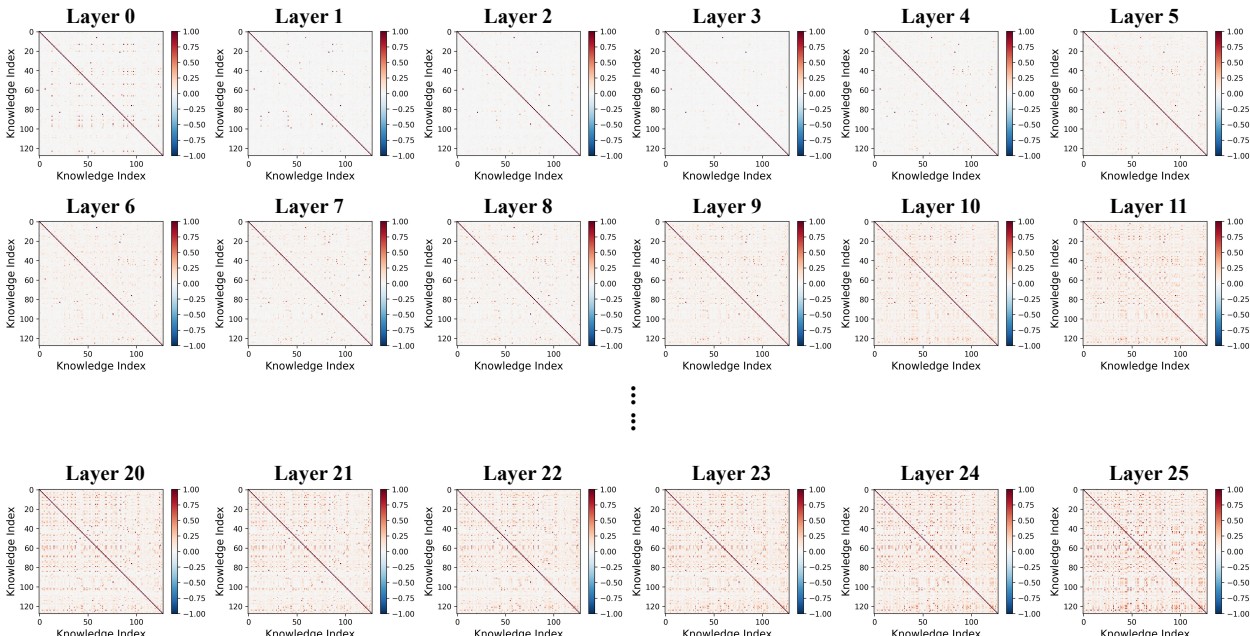

*Figure 15.* Superposition of activations on Gemma2-2B.

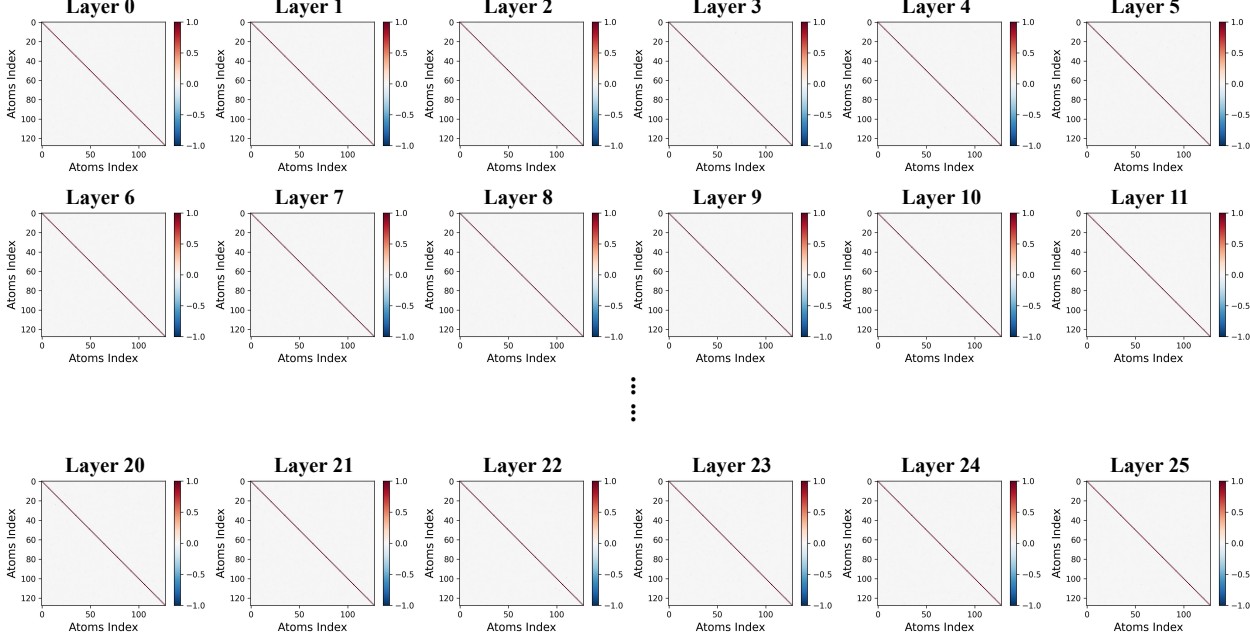

*Figure 16.* Solving superposition on Gemma2-2B.

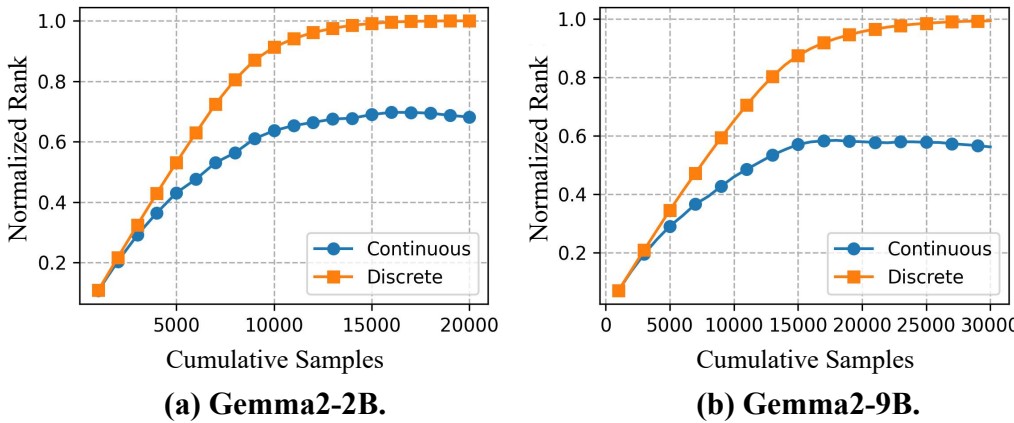

**(a) Gemma2-2B.**         **(b) Gemma2-9B.**

*Figure 17.* Cumulative normalized rank of (a) Gemma2-2B and (b) Gemma2-9B. Each data point corresponds to the ratio between the rank of the accumulated activation matrix (formed by stacking samples up to that point) and the total dimensionality (i.e., the theoretical maximum rank). Here we illustrate this for randomly selected early layers of Gemma2-2B and Gemma2-9B.

*Table 2.* Comparison of reconstruction quality and sparsity under different training data sources.

| Training Data Source | $R^2$ | $L_0$ |
|---|---|---|
| General corpora (Wikipedia) | 99.84% | 9.36 |
| Complex Reasoning (MATH500) | 99.86% | 9.56 |
| Knowledge Atomization (WikiData) | 99.89% | 8.32 |

## C. Atoms of LLMs

### C.1. Training Paradigm

We train threshold-activated sparse autoencoders (TSAEs) on activations extracted by entity knowledge, a setting we term the knowledge atomization task. Unlike the common practice of training on activations from natural corpora, this formulation enables precise control and quantification of data scale, facilitating scalable and systematic study across model and dataset scales. Moreover, entity-induced activations exhibit higher normalized rank (Fig. 17), spanning a broader set of representational dimensions and thus providing richer information.

We also conduct task-level ablations. Specifically, we train TSAEs on activations extracted from natural text (Wikipedia (In, 2001)) and complex mathematical reasoning data (MATH500 (Hendrycks et al., 2021)), following the same training pipeline as knowledge atomization while matching model and data scales. As shown in Tab. 2, representations obtained from different sources yield consistent results across language models, demonstrating the generality of our findings.

### C.2. Data Collection

In § 4.1 and § 4.3, we use the WikiData dataset (Vrandečić & Krötzsch, 2014), while in § 4.2 and § 4.4 we adopt the CounterFact dataset (Meng et al., 2022).

Specifically, we collect activations from every layer of Gemma2-2B, Gemma2-9B, and Llama3.1-8B using the subject entities in the corresponding datasets (e.g., "Danielle Darrieux," "Edwin of Northumbria," and "Toko Yasuda").

Activations are collected in a uniform manner: each subject name is used as a prompt, and hooks record activations at the final token of the subject mention, a position previously identified as critical for knowledge recall in language models (Meng et al., 2022). The resulting activations are aggregated as static training data.

## C.3. Training Details

We employ single-layer SAEs with threshold activation, denoted as $f : \boldsymbol{x} \mapsto \hat{\boldsymbol{x}} = W_{\mathrm{dec}}\, \sigma(W_{\mathrm{enc}}\boldsymbol{x})$, and train it by minimizing a joint reconstruction–sparsity objective

$$\mathcal{L}(\boldsymbol{x}) = \underbrace{\|\boldsymbol{x} - \hat{\boldsymbol{x}}\|_2^2}_{\mathcal{L}_{\mathrm{reconstruct}}} + \lambda \underbrace{\|\sigma(\boldsymbol{z})\|_1}_{\mathcal{L}_{\mathrm{sparsity}}}, \tag{C.1}$$

where $\boldsymbol{z} = W_{\mathrm{enc}}\boldsymbol{x}$, and $\sigma$ is coordinate-wise JumpReLU activation (Rajamanoharan et al., 2024b),

$$\bigl(\sigma(\boldsymbol{z})\bigr)_i = \begin{cases} 0, & z_i < \tau_i, \\ z_i, & z_i \geq \tau_i, \end{cases} \quad \boldsymbol{\tau} = (\tau_i)_i. \tag{C.2}$$

The key hyperparameters are the sparsity coefficient $\lambda$ in the loss function (Eq. C.1) and the threshold initialization. We fix $\lambda = 0.1$ (later shown to be insensitive) and initialize the threshold at 0.001 (or 0.0001), which provides a good trade-off between training efficiency and effectiveness: smaller initial thresholds facilitate satisfying the support-separation condition but substantially increase training time, whereas 0.001 serves as a stable and reliable default in our experiments. During training, we employ the straight-through estimator (Rajamanoharan et al., 2024b) to approximate gradients at the non-differentiable threshold.

We select the final model as the checkpoint on the Pareto front that optimally balances reconstruction error and sparsity. Fig. 18 illustrates the Pareto front for Gemma2-2B.

The specific computational cost is as follows:

- Gemma2-2B (per layer): $\sim$24 GPU-hours on RTX 3090-24G (on average);

- Gemma2-9B (per layer): $\sim$56 GPU-hours on A100-80G (on average);

- Llama3.1-8B (per layer): $\sim$58 GPU-hours on A100-80G (on average);

- Largest TSAE trained in this work (Fig. 5, top-right): $\sim$135 GPU-hours on A100-80G.

A minor training issue was observed in layers 30 and 31 of Llama 3.1-8B, where unusually large activations caused optimization to fail. Consequently, these layers are omitted from the reported results. This behavior is likely related to their proximity to the output, where activations may drive next-token prediction rather than encode entity-specific information. Further analysis indicates that the failure is mainly due to abnormally high data variance in these layers; normalizing the training data by its distributional standard deviation, followed by a simple reparameterization of the trained model, largely resolves the issue. By contrast, Gemma 2-2B and Gemma 2-9B did not exhibit this problem, possibly because their extensive use of RMSNorm mitigates such activation outliers.

## C.4. Baseline Details

The primary baselines used in this work are GemmaScope and LlamaScope. GemmaScope provides SAEs of widths 16k and 65k trained on the MLP layers of Gemma2-2B, as well as SAEs of widths 16k and 131k trained on the MLP layers of Gemma2-9B. LlamaScope offers SAEs with expansion factors of 8× and 32× trained on the MLP layers of Llama3.1-8B. Both GemmaScope and LlamaScope are widely regarded as open-source tools for feature extraction.

It is important to emphasize that these models are trained on activations derived from continuous text corpora. We use them as baselines not to demonstrate superior performance of our SAEs, but to highlight that feature-based reconstruction of raw activations remains unreliable in practice, whereas our results show that internal representations of language models can be reconstructed with high fidelity.

## C.5. Evaluation Details

To practically assess the stability of identified representational units, we introduce quantile-based statistics that correspond to the prior conditions in Theorem 3.9. Specifically, we define two statistics:

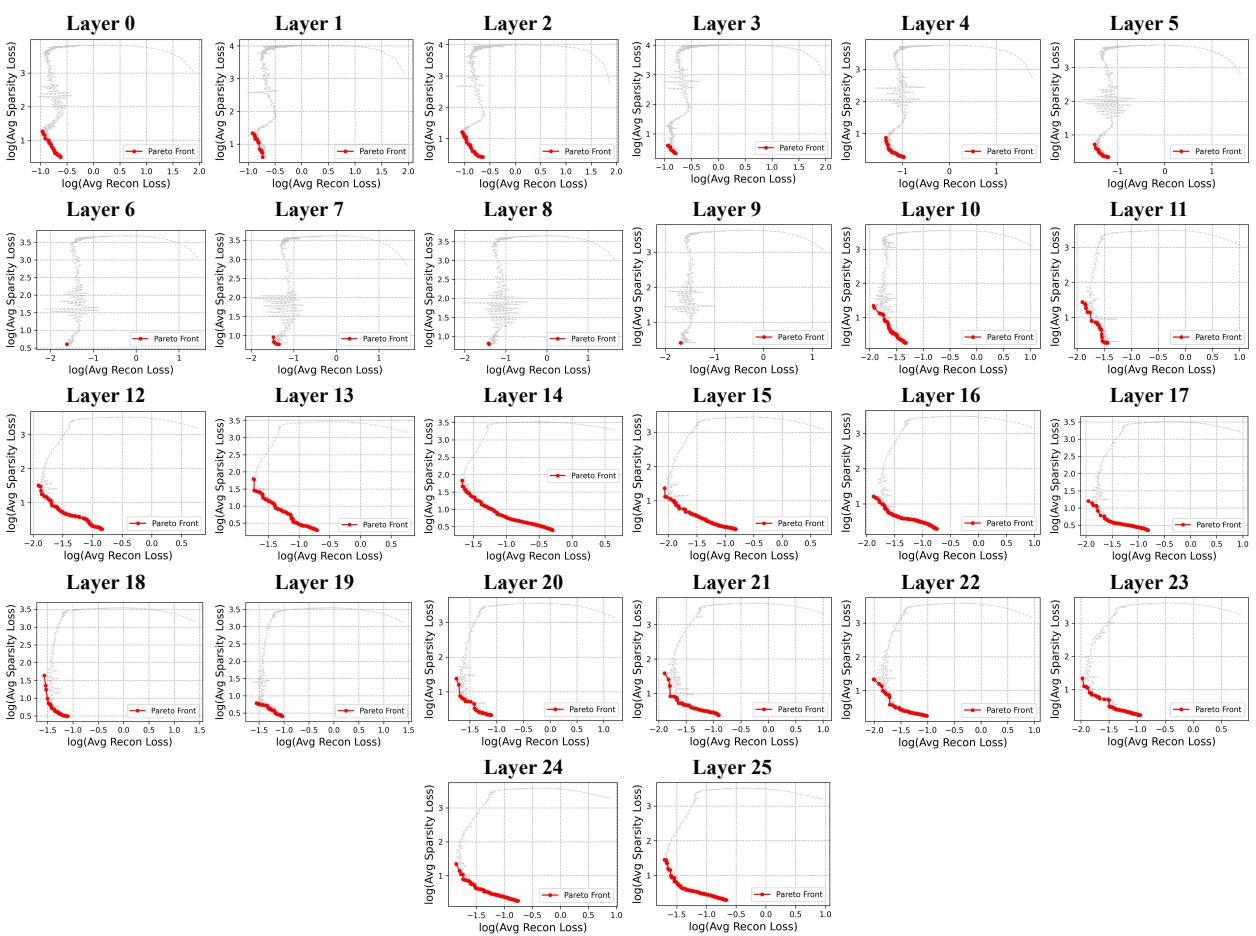

*Figure 18.* Pareto front during training on Gemma2-2B.

- **Quantile sparsity** $K_q$: The quantile of sparsity $K_q$ is defined as

$$K_q = \inf \left\{ k \in \mathbb{N} : \mathbb{P}_{\boldsymbol{\delta} \sim \mathcal{P}_\Delta} \left( K \leq k \right) \geq q \right\}, \tag{C.3}$$

where $\boldsymbol{\delta}$ is a coefficient vector sampled from the distribution $\mathcal{P}_\Delta$, and the random variable $K := \|\boldsymbol{\delta}\|_0$ represents the sparsity of the sampled coefficient vector. In simple terms, the quantile sparsity $K_q$ indicates that at least $q$ of the samples have sparsity no greater than $K_q$.

- **Quantile coherence** $\mu_q$: Similarly, the quantile of coherence $\mu_q$ is defined as

$$\mu_q = \inf \left\{ \mu \geq 0 : \mathbb{P}_{(\mathcal{I},\mathcal{J})} \left( C \leq \mu | \tilde{D} \right) \geq q \right\}, \tag{C.4}$$

where $(\mathcal{I}, \mathcal{J})$ is uniformly sampled from all unordered pairs of indices ($\mathcal{I} \neq \mathcal{J}$), and the random variable $C := |\langle \tilde{\boldsymbol{d}}_\mathcal{I}, \tilde{\boldsymbol{d}}_\mathcal{J} \rangle|$ represents the coherence between two atoms. In simple terms, this means that the probability of randomly selecting a pair of different atoms with coherence no greater than $\mu_q$ is at least $q$.

Based on these definitions, for the supports of most samples, if the condition $\mu_q < \frac{1}{2K_q - 1}$ holds, we can conclude that at least $q$ proportion of the samples satisfy the sufficient conditions for uniqueness and recoverability.

To determine the maximal quantile $q^*$ satisfying the theoretical criterion, we perform a binary search over the interval $[0, 0.999999]$ for the quantile parameter $\alpha$. At each iteration we compute the linear quantiles

$$\mu_\alpha := \mathrm{Quantile}(\{\mu\}, \alpha), \quad K_\alpha := \mathrm{Quantile}(\{K\}, \alpha), \tag{C.5}$$

and test whether $\mu_\alpha < \frac{1}{2K_\alpha - 1}$ holds. If the condition is satisfied, the lower bound of the search interval is updated to $\alpha$; otherwise the upper bound is reduced. Upon convergence, the maximal $\alpha$ obtained is taken as the desired quantile $q^*$, together with the corresponding values of $\mu_\alpha$ and $K_\alpha$.

Note that verifying Theorem 3.9 requires the equality $\tilde{D}\boldsymbol{x} = \tilde{\boldsymbol{m}}$. However, as shown in Fig. 4, features generally fail to achieve reliable reconstruction, so the quantile $q$ obtained from the condition $\mu_q < \frac{1}{2K_q - 1}$ serves only as an ideal upper bound. In contrast, the learned atoms satisfy reliable reconstruction, and 99.85% of atoms meet $\mu_q < \frac{1}{2K_q - 1}$ on average, confirming their favorable properties.

For further detail, Tabs. 3-5 report the corresponding values of $R^2$ and $q^*$ for identified units of Gemma2-2B, Gemma2-9B and Llama3.1-8B. In all three models, we primarily use TSAEs with JumpReLU activations (Erichson et al., 2019; Rajamanoharan et al., 2024b) to identify atoms. For comparison, on Gemma2-2B we also train standard SAEs with ReLU activations (Cunningham et al., 2023) and find that they fail to identify units that satisfy the criteria of ideal atoms as fundamental representational units. Moreover, control experiments with ReLU SAEs of varying capacities show that increasing capacity does not improve performance (Tab. 6), which is also consistent with our theoretical expectation.

### C.6. Experimental Analysis

Notably, the training process is largely insensitive to hyperparameters: using sparsity coefficients $\lambda \in \{0.01, 0.1, 1\}$ yields nearly identical learning curves (Fig. 19), indicating strong robustness. This suggests that high-fidelity reconstruction primarily reflects the intrinsic sparsifiability of the representations, rather than careful hyperparameter tuning.

The encoder and decoder of SAEs converge to alignment under atomic inner product, namely parameterization of $W_{dec} = D$ and $W_{enc} = D^\top \tilde{S}$, consistent with Theorem 3.11, as shown in Fig. 20 for Gemma2-2B. Results for Gemma2-9B and Llama3.1-8B are omitted due to the file-size limit. Full results are provided in the arXiv version and are also available upon request.

By Definition 3.6, atoms must satisfy approximate orthogonality under the normalized atomic inner product (NAIP), ensuring their mutual distinguishability. The NAIP among all atoms can be computed by directly evaluating the matrix $G = \tilde{D}^\top \tilde{D}$, with a more practical procedure, similar to Corollary 3.3, given by

$$G = \frac{D^\top S D}{\sqrt{\mathrm{diag}(D^\top S D)} \times \sqrt{\mathrm{diag}(D^\top S D)}}, \tag{C.6}$$

*Table 3.* Faithfulness and stability across layers on Gemma2-2B.

| Layer | TSAEs with JumpReLU | | SAEs with ReLU | |
|---|---|---|---|---|
| | $R^2$ | $q^*$ | $R^2$ | $q^*$ |
| 0 | 0.9986 | 0.9974 | - | - |
| 1 | 0.9984 | 0.9978 | - | - |
| 2 | 0.9987 | 0.9978 | - | - |
| 3 | 0.9992 | 0.9988 | - | - |
| 4 | 0.9994 | 0.9991 | - | - |
| 5 | 0.9996 | 0.9983 | 0.9680 | 0.8263 |
| 6 | 0.9995 | 0.9983 | 0.9510 | 0.6740 |
| 7 | 0.9993 | 0.9994 | 0.9624 | 0.6712 |
| 8 | 0.9996 | 0.9976 | 0.9650 | 0.6437 |
| 9 | 0.9995 | 0.9987 | 0.9383 | 0.5648 |
| 10 | 0.9992 | 0.9983 | 0.9133 | 0.4662 |
| 11 | 0.9992 | 0.9979 | 0.9179 | 0.4366 |
| 12 | 0.9991 | 0.9911 | 0.9129 | 0.4516 |
| 13 | 0.9973 | 0.9960 | 0.9104 | 0.4167 |
| 14 | 0.9989 | 0.9993 | - | - |
| 15 | 0.9988 | 0.9989 | - | - |
| 16 | 0.9992 | 0.9972 | - | - |
| 17 | 0.9994 | 0.9991 | - | - |
| 18 | 0.9996 | 0.9945 | - | - |
| 19 | 0.9997 | 0.9980 | - | - |
| 20 | 0.9989 | 0.9989 | - | - |
| 21 | 0.9997 | 0.9964 | - | - |
| 22 | 0.9995 | 0.9922 | - | - |
| 23 | 0.9994 | 0.9979 | - | - |
| 24 | 0.9994 | 0.9982 | - | - |
| 25 | 0.9993 | 0.9943 | - | - |

*Table 4.* Faithfulness and stability across layers on Gemma2-9B.

| | TSAEs with JumpReLU | |
|---|---|---|
| Layer | $R^2$ | $q^*$ |
| 0 | 0.9996 | 0.9915 |
| 1 | 0.9993 | 0.9992 |
| 2 | 0.9995 | 0.9981 |
| 3 | 0.9996 | 0.9975 |
| 4 | 0.9996 | 0.9985 |
| 5 | 0.9997 | 0.9999 |
| 6 | 0.9993 | 0.9961 |
| 7 | 0.9996 | 0.9995 |
| 8 | 0.9996 | 0.9996 |
| 9 | 0.9996 | 0.9997 |
| 10 | 0.9997 | 0.9997 |
| 11 | 0.9996 | 0.9994 |
| 12 | 0.9994 | 0.9996 |
| 13 | 0.9993 | 0.9991 |
| 14 | 0.9989 | 0.9991 |
| 15 | 0.9992 | 0.9997 |
| 16 | 0.9990 | 0.9997 |
| 17 | 0.9994 | 0.9996 |
| 18 | 0.9995 | 0.9995 |
| 19 | 0.9990 | 0.9996 |
| 20 | 0.9991 | 0.9995 |
| 21 | 0.9991 | 0.9994 |
| 22 | 0.9992 | 0.9993 |
| 23 | 0.9993 | 0.9991 |
| 24 | 0.9993 | 0.9990 |
| 25 | 0.9994 | 0.9988 |
| 26 | 0.9964 | 0.9974 |
| 27 | 0.9997 | 0.9980 |
| 28 | 0.9994 | 0.9997 |
| 29 | 0.9993 | 0.9993 |
| 30 | 0.9997 | 0.9982 |
| 31 | 0.9995 | 0.9982 |
| 32 | 0.9996 | 0.9982 |
| 33 | 0.9998 | 0.9986 |
| 34 | 0.9998 | 0.9987 |
| 35 | 0.9997 | 0.9997 |
| 36 | 0.9997 | 0.9991 |
| 37 | 0.9995 | 0.9992 |
| 38 | 0.9993 | 0.9996 |
| 39 | 0.9990 | 0.9998 |
| 40 | 0.9988 | 0.9999 |
| 41 | 0.9995 | 0.9951 |

*Table 5.* Faithfulness and stability across layers on Llama3.1-8B.

| Layer | TSAEs with JumpReLU | |
| | $R^2$ | $q^*$ |
|---|---|---|
| 0 | 0.9985 | 0.9968 |
| 1 | 0.9998 | 0.9996 |
| 2 | 0.9930 | 0.9998 |
| 3 | 0.9945 | 0.9999 |
| 4 | 0.9992 | 0.9999 |
| 5 | 0.9992 | 0.9998 |
| 6 | 0.9971 | 0.9999 |
| 7 | 0.9961 | 0.9999 |
| 8 | 0.9992 | 0.9999 |
| 9 | 0.9988 | 0.9999 |
| 10 | 0.9989 | 0.9998 |
| 11 | 0.9987 | 0.9999 |
| 12 | 0.9993 | 0.9997 |
| 13 | 0.9970 | 0.9999 |
| 14 | 0.9992 | 0.9999 |
| 15 | 0.9986 | 0.9999 |
| 16 | 0.9989 | 0.9999 |
| 17 | 0.9992 | 0.9998 |
| 18 | 0.9992 | 0.9994 |
| 19 | 0.9993 | 0.9998 |
| 20 | 0.9991 | 0.9998 |
| 21 | 0.9993 | 0.9996 |
| 22 | 0.9997 | 0.9992 |
| 23 | 0.9995 | 0.9996 |
| 24 | 0.9996 | 0.9993 |
| 25 | 0.9990 | 0.9999 |
| 26 | 0.9995 | 0.9997 |
| 27 | 0.9993 | 0.9999 |
| 28 | 0.9982 | 0.9993 |
| 29 | 0.9979 | 0.9940 |

*Table 6.* Performance of ReLU SAEs with different capacities on Gemma2-2B at layer 10.

| Capacity | $4\times$ | $5\times$ | $6\times$ | $7\times$ |
|---|---|---|---|---|
| $R^2$ | 0.9183 | 0.9173 | 0.9150 | 0.9143 |
| $q^*$ | 0.4768 | 0.4655 | 0.4514 | 0.4532 |

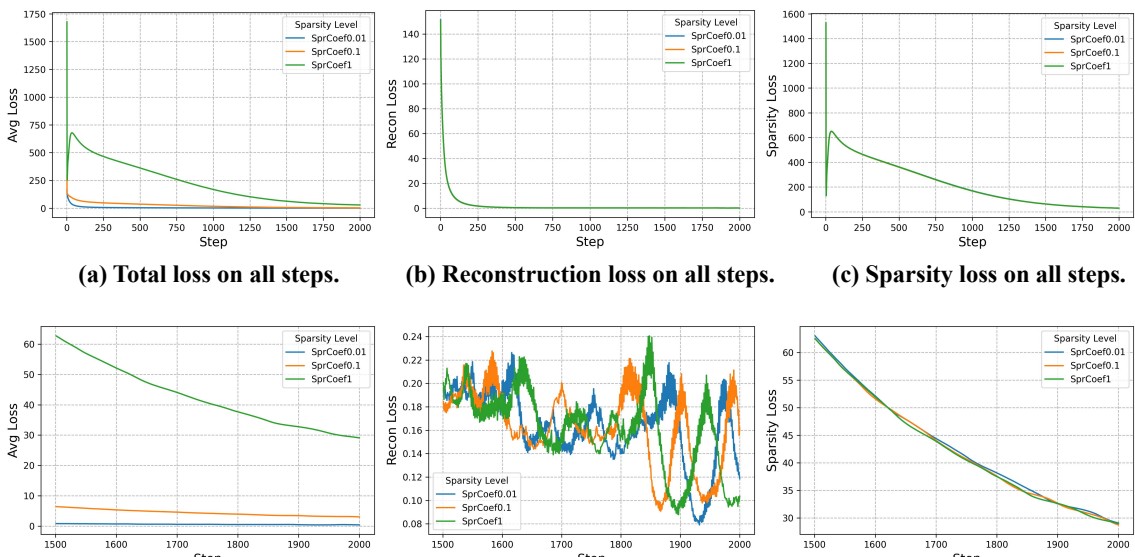

(a) **Total loss on all steps.**    (b) **Reconstruction loss on all steps.**    (c) **Sparsity loss on all steps.**

(d) **Total loss on last 500 steps.** (e) **Reconstruction loss on last 500 steps.** (f) **Sparsity loss on last 500 steps.**

*Figure 19.* Training loss is robust to hyperparameter selection on $\lambda$, maintaining stable performance across different configurations.

where $S = (DD^\top)^{-1}$, $\mathrm{diag}(D^\top SD)$ denotes the diagonal of $D^\top SD$, $\sqrt{\mathrm{diag}(D^\top SD)}$ denotes its element-wise square root, and $\times$ indicates the outer product. If the vectors learned by the SAEs exhibit atomicity, the off-diagonal elements $G_{ij} = \langle \tilde{d}_i, \tilde{d}_j \rangle$ should cluster near zero with very small variance, demonstrating approximate orthogonality, while the diagonal entries are normalized.

As shown in Fig. 21, across all layers of Gemma2-2B, the off-diagonal elements of the matrices $D$ are tightly concentrated near zero, closely matching the theoretical Dirac delta distribution. This accords with Definition 3.5: although strict orthogonality is unattainable, sparsity drive convergence to approximately orthogonal atoms. Results for Gemma2-9B and Llama3.1-8B are omitted due to the file-size limit. Full results are provided in the arXiv version and are also available upon request.

### C.7. Monosemanticity Evaluation

To evaluate the monosemanticity of the representational units, we adopt LLM-as-a-judge.

Specifically, to ensure diversity, we first manually select a set of heterogeneous entities (e.g., "United Kingdom", "Google Maps", "Suzuki GSX-R750", "Windows Vista", "Intel 80286", "Beijing", "Hawaii", "Tim Duncan", "Microsoft Word", "Vladimir Putin", "Apple Watch", "Chrome OS"). We then collect the representational units activated by these entities and aggregate them into a candidate pool. From this pool, we randomly sample ten units per selected layer: Gemma2-2B (layers 0, 5, 10, 15, 20, 25), Gemma2-9B (layers 0, 7, 14, 21, 28, 35, 41), and Llama3.1-8B (layers 0, 6, 12, 18, 24, 29). For each sampled unit, we retrieve all entities that activate it and use GPT-5.2 to assess its monosemanticity. The evaluation prompt is provided below:

> You are given a list of entities enclosed in square brackets [ ].
> Inside the brackets, each entity is separated by a semicolon (;).
> Your task is to analyze the entities and determine how many of them belong to the same semantic category (i.e., refer to

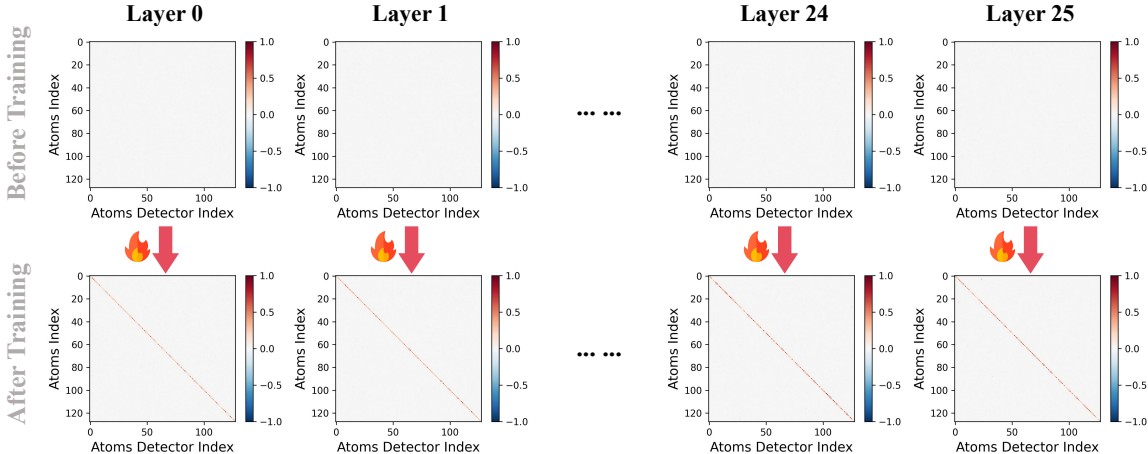

*Figure 20.* Spontaneous alignment between the encoder and decoder during training on Gemma2-2B.

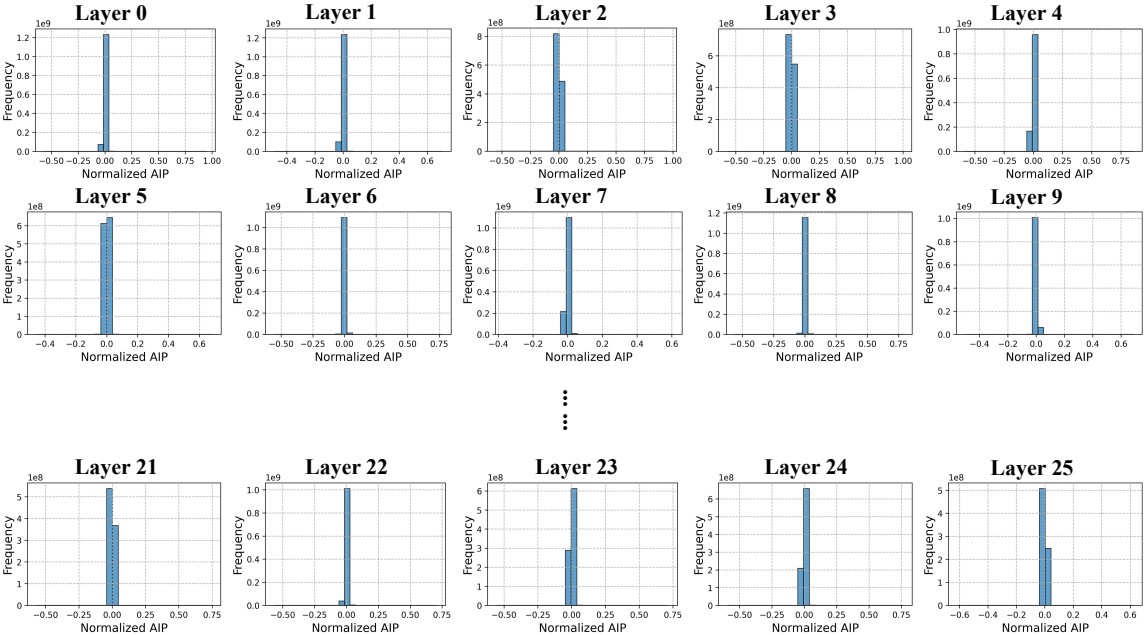

*Figure 21.* NAIP distribution of atoms across all layers of the Gemma2-2B.

the same type of real-world concept).
Important instructions:
- You should identify the largest group of entities that belong to the same category.
- Only count entities that clearly belong to the same category.
- Your answer must be a single integer.
You must provide your final answer strictly inside a box using the following format:
\box{NUMBER}
Here is the list of entities: [{entities}]

We then compute, for each representational unit, the proportion of activated entities that are monosemantic, and report the mean and standard error of the mean (SEM) over units sampled at selected layers. Full results are shown in Fig. 22, revealing that monosemanticity increases with model scale and is generally higher in deeper layers than in shallower ones. Aggregating across layers within each model yields the results presented in Fig. 6.

For example, in Gemma2-9B, an atom (ID 11346) in layer 28 is activated by entities including "Honolulu," "aloha," "Mufi Hannemann," "Kirk Caldwell," "Hawaii," "Hawaiian Islands," "Mauna Kea," "USS Honolulu," and "Aloha Stadium." Notably, Mufi Hannemann was born in Honolulu and served as its mayor; Kirk Caldwell is a former Hawaii state representative and former mayor of Honolulu; and Mauna Kea is a volcano in the Hawaiian Islands. This example demonstrates that the atom consistently captures a semantically coherent "Hawaii–Honolulu" concept region, exhibiting clear monosemanticity.

Furthermore, we analyze atoms activated by "Beijing" in layers 1–6 of Gemma2-2B, and examine, at each layer, all entities that activate these atoms to characterize their corresponding concept regions (Tabs. 7–12).

*Table 7.* Entities grouped by atoms ID for *Beijing* on layer 1 of Gemma2-2B.

| Atoms ID | Entities |
| --- | --- |
| 15264 | Beijing, Seoul, 1 Maccabees, Ulysses Dove |
| 15982 | Beijing, Siikainen, 36 China Town, Jim Allchin |
| 23987 | Beijing, Swann Memorial Fountain, Charles Chilton, Otto Neurath |
| 31322 | Shanghai, Beijing |
| 35951 | Beijing, Russia, Arkansas, Paris |
| 36035 | Beijing, Meiert Avis, Aviation Industry Corporation of China |

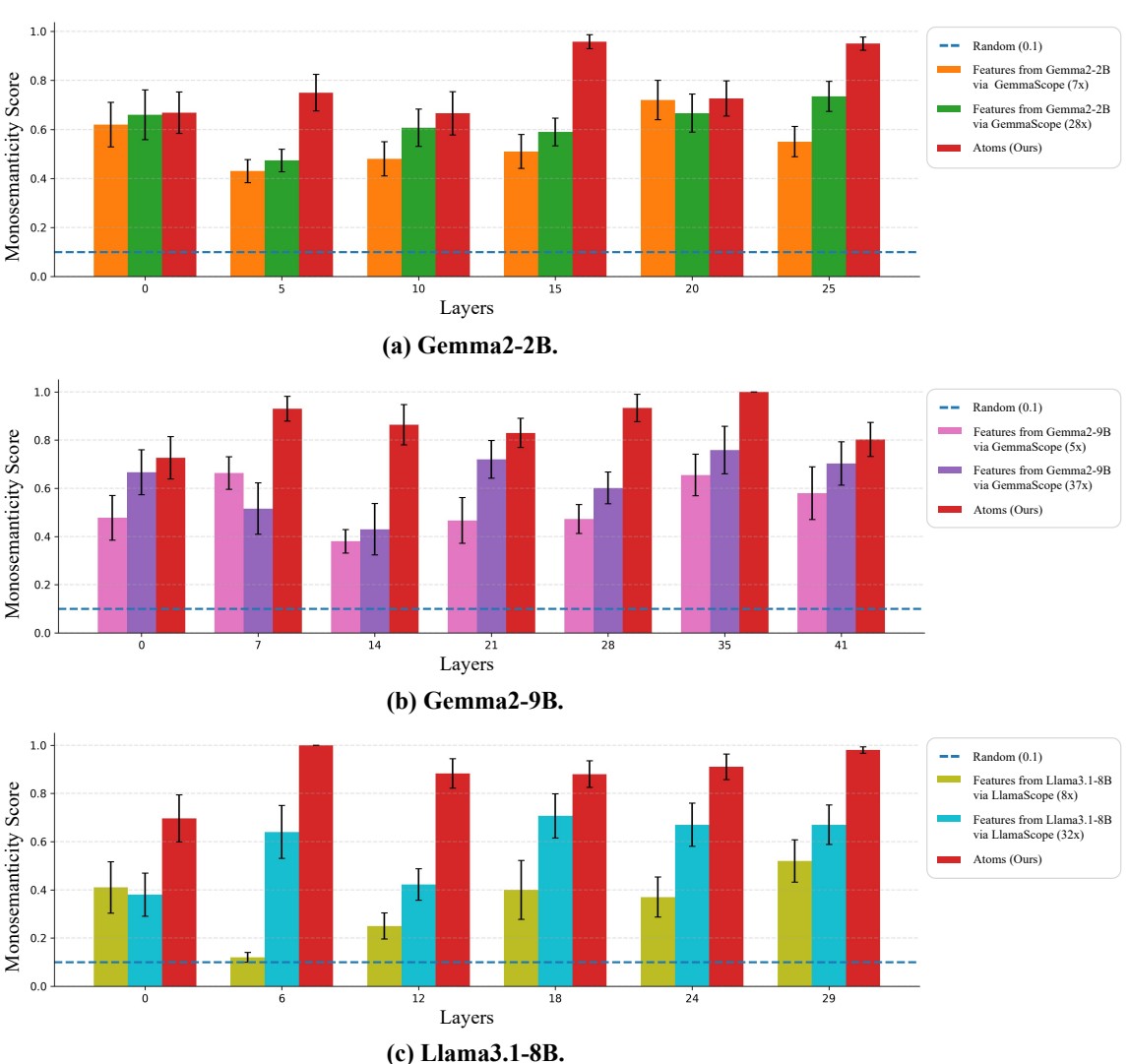

(a) Gemma2-2B.

(b) Gemma2-9B.

(c) Llama3.1-8B.

*Figure 22.* Monosemanticity scores of representational units across models and layers, using GPT-5.2 with manual verification. The blue dashed line indicates random-guess performance (0.1).

*Table 8.* Entities grouped by atoms ID for *Beijing* on layer 2 of Gemma2-2B.

| Atoms ID | Entities |
|---|---|
| 620 | Shanghai, Beijing, Hanoi, Tokyo, Adam Maida |
| 6258 | Beijing, Majorca, Thailand, Greg Dyke |
| 7540 | 1300 Oslo, Beijing, Miami Horror, Lille |
| 10761 | Moscow, Beijing, Canberra, Pyongyang |
| 11519 | Karl Polanyi, Beijing, Cevdet Sunay, Mary Gaunt, Cyd Hayman, Les diamants de la couronne |
| 13418 | Beijing, Tarnobrzeg Voivodeship, Yakuza, Longs Peak, Jeep Wrangler |
| 15585 | Beijing, Ivan Koloff, Olinto Cristina |
| 22622 | Shanghai, Cleveland, Beijing, Delhi, Saint Lucia, St Lucia, Venice |
| 26002 | Beijing, Alte Oper, Intimate Stories, Seventeen, Five Star Krishna |
| 27116 | Ankara, Mandarin Oriental, Bangkok, Cairo, Beijing, Dublin, Jakarta, Amsterdam, Bratislava, Toronto, Sydney, Edinburgh, London, Honolulu, Auckland, Bali, Tokyo, Manila, Queens Gardens, Brisbane, Budapest, Montreal, Perth, Kolkata, Dubai, Melbourne, Copenhagen, Nairobi, Bangkok, Bangalore |

*Table 9.* Entities grouped by atoms ID for *Beijing* on layer 3 of Gemma2-2B.

| Atoms ID | Entities |
|---|---|
| 9444 | Shanghai, Beijing |
| 24724 | Moscow, Beijing, Russia |
| 30463 | Beijing, Thailand |
| 32854 | Beijing, Madrid, Mariano Gonzalvo |

*Table 10.* Entities grouped by atoms ID for *Beijing* on layer 4 of Gemma2-2B.

| Atoms ID | Entities |
|---|---|
| 1578 | Beijing, Cadbury |
| 11098 | Beijing, Jakarta |
| 11158 | Beijing |
| 15601 | Oslo, Moscow, Stockholm, Berlin, Athens, Helsinki, Beijing, Vienna, Geneva, Amsterdam, Seoul, Prague, Madrid, London, Warsaw, Kyoto, Naples, Tokyo, Budapest, Paris, Rome, Bangkok |
| 25755 | Stockholm, Helsinki, Beijing, Minneapolis, Minecraft, Copenhagen, Nairobi |
| 33322 | Shanghai, Beijing, Guangzhou, Macau, Hong Kong, Chongqing, Shenzhen, Wuhan |

*Table 11.* Entities grouped by atoms ID for *Beijing* on layer 5 of Gemma2-2B.

| Atoms ID | Entities |
|---|---|
| 11453 | Beijing, The Great Citizen |
| 12661 | Beijing, Holycross-Ballycahill GAA |
| 19018 | Beijing, Registro, 4th of August Regime, Witnesses |
| 23750 | Moscow, Ankara, Beijing, Jakarta, Madrid |

*Table 12.* Entities grouped by atoms ID for *Beijing* on layer 6 of Gemma2-2B.

| Atoms ID | Entities |
|---|---|
| 7533 | Johannesburg, Shanghai, Beijing, Colombo, Prafulla Chandra Ghosh |
| 16414 | Shenyang, Shanghai, Beijing, Guangzhou, Yangtze, Google China, Taobao, Tianjin, Chongqing, National Development and Reform Commission, Shenzhen, Qing dynasty, Aviation Industry Corporation of China, Qzone, Youku, Wuhan, People's Republic of China |
| 22386 | Beijing |
| 33958 | Carol Zhao, Shenyang, Shanghai, Beijing, Guangzhou, Seoul, Yangtze, Macau, Hanoi, Taipei, Hong Kong, Kaohsiung, South Korea, Busan, United States Army Military Government in Korea, Tianjin, Pyongyang, Incheon, Chongqing, Vietnam, Dennis Hwang, Shenzhen, Daejeon, North Korea, Wuhan |

