# OpenReview forum: "Towards Atoms of Large Language Models"
_ICML.cc/2026/Conference — ICML 2026 regular_

### Official Review · Reviewer_iX87 · 2026-03-11

**Soundness:** 2
**Presentation:** 2
**Significance:** 2
**Originality:** 2
**Overall Recommendation:** 2
**Confidence:** 4

**Summary:**

The paper proposes to use SAE to extract atoms. These are then used to define an inner product, which they call AIP. The inner product is used to compare embeddings in LLMs. They propose faithfulness and stability as two desirable properties of said atoms. They conduct experiments on LLMs to motivate this construction and also justify it theoretically.

**Compliance With Llm Reviewing Policy:**

Affirmed.

**Final Justification:**

The original paper's presentation was not sufficiently clear. During the rebuttal, the authors clarified the points of confusion and promised to modify the paper accordingly. But I can not see the updated paper and therefore can't say whether the clarity issues are now resolved.

The original paper lacked citations of prior works. The authors added some new citations during the rebuttal, but they still miss more citations to other relevant works. It shouldn't be hard to cite relevant literature appropriately, given that there aren't that many works to begin with, studying AI through the lens of compressed sensing.

**Key Questions For Authors:**

- “When representation angles are computed under the EIP, the centroid of the angular distribution deviates markedly from 90◦ , revealing a representation shift” — did you try subtracting a global average from all the atoms in D prior to computing the cosine similarity?
- For figure 3, and in fact in general, how did you find the matrix D used to construct the metric S?
- What’s the intuition/reason behind defining q*?
- Isn’t the criterion for success of mu<1/(2k-1) generally very loose?
- How do you define and compute K_q for an LLM?
- How do you compute the x and y values and also color of each cell in figure 5?
- Why are the results in table 1 so high? Could you specify what is the coherence, sparsity, dimensions, etc.?

**Limitations:**

Yes

**Strengths And Weaknesses:**

- Figure 1 is not very clear. What is mu_q, K_q, q*, etc.?
- Throughout the introduction it’s unclear what’s the definition of q*.
- The authors don’t cite papers that originally proposed definition 3.5 (see literature on mutual coherence). They also don’t cite papers that used mutual coherence for theoretical analysis of deep learning. They also don’t cite the papers that originally proved theorem 3.8, theorem 3.9, etc.
- The atomic inner product and atoms are discussed throughout the introduction but are not defined until pages 3 and 4 which makes reading the introduction pretty confusing. The definition is also confusing and possibly circular. In the introduction the authors refer to atoms in the following way: first define the AIP, and based on this define atoms as satisfying three properties: representability, sparsity, and separability (atoms are approx. orthogonal under the AIP). The AIP is then defined in the following way: take the set of atoms D = {d_i}_i, then the atomic inner product <, > is an inner product on span(D) under which the atoms are orthogonal. The atoms are then defined as being a certain collection of vectors under which this inner product is almost orthogonal (and satisfies a couple other properties). The framing here is confusing. To actually compute the AIP, you need to have the matrix \tilde{S}. To have this matrix you need the atoms. The authors say they can estimate this matrix in practice by taking E[kk^T]^{-1}, where k is a vector of activations. But why can you do this?
- The abstract says that "neurons are faithful (R^2 = 1) but unstable (q* = 0.5%), while features are more stable (q* = 68.2%) but unfaithful (R^2 = 48.8%)." The details for this are pretty buried in the text. The authors say its an average of all the layers over three LLMs they tested which is fair I guess but I found it pretty confusing at first. The authors also refer to figure 4 for the numbers, but the figure is hard to read and the caption isn’t illuminating. It is too small and scaling of axis is too tight to distinguish anything. You can infer that the outer numbers on the circles are the layers of the models, but I don’t see why a circle is a natural way to represent model layers.
- Language throughout is vague and/or hyperbolic: abstract says "thereby grounding atom theory", grounding means what here? And why does this alone ground atom theory? In the introduction authors say superposition “raises doubts about [neurons’] validity for analysis”, this is vague and are they invalid? In the conclusion they say “Overall, these results elevate interpretability from heuristic analysis to a principled theory of FRUs in LLMs.” This makes it sound like this paper has just established interpretability as a legitimate field and everyone else was just making it up.
- The related work section seems kind of underdeveloped perhaps. Authors don’t mention if there have been other attempts to study anything other than neurons and features.
- Presumably the goal of this work is that it adds some transparency to LLM interpretability. The authors even say these results “elevate interpretability from heuristic analysis to a principled theory”. But they do not apply the theory to any particularly interesting interpretability results. I think the definition of atoms needs to be justified as useful. There should be an application where atoms revealed something that neurons or features couldn’t

---

> ### Author Rebuttal · Authors · 2026-03-30
>
> Sincerely thank you for your valuable feedback. We respond to the comments as follows.
> 1. **On subtracting the global mean (Q1).**
>
>    Our experiments show that when the angle distribution is unimodal, this method shifts the mean toward $90^\circ$, consistent with the reviewer’s intuition.
>
>    However, we observe that such subtraction introduces two issues: (i) **Unstable distribution structure**: it flattens the angle distribution (increasing variance), obscuring discriminative features (e.g., heavy tails) and reducing cross-layer comparability; (ii) **limited handling of complex distribution**: real distributions are often non-unimodal (e.g., bimodal in Gemma2-2B layers 9–14 and 17), where mean subtraction can introduce additional distortions.
>
>    In contrast, we introduce AIP, which aligns with the underlying representation structure (see L130–159 left). Under AIP, angle distributions show consistent patterns across models and layers, providing a unified geometric view.
>
>    **In summary, AIP is theoretically grounded (L130–159 left) and consistently validated across model families ($\S$4.1), whereas mean subtraction is a heuristic method, lacking theoretical support and potentially yield misleading conclusions.**
>
>    We will add comparative experiments in the revision to further clarify the differences.
>
> 2. **On the $\tilde S$ used to compute AIP (Q2&W4).**
>
>    Please refer to our response to Reviewer VPEj (Point 1).
>
> 3. **On the intuition behind defining $q^*$ (Q3).**
>
>    The core intuition is to identify **fundamental representational units (atoms)** that are detectable or manipulable without interfering with others (**distinguishability**) (see L129-135 left). Ideally, this corresponds to **strict orthogonality**, but in an $H$-dimensional space, at most $H$ such vectors exist, which limits expressiveness.
>
>    Compressed sensing shows strict orthogonality is unnecessary, and distinguishability can be preserved via sparsity and approximate orthogonality. The metric $q^*$ quantifies this in the generalized setting (see L223–232 right).
>
> 4. **On the success criterion (Q4).**
>
>    This condition is not a heuristic ‘loose’ or ‘strict’ choice, but is rigorously derived from compressed sensing theory. Once satisfied, it provides a clear theoretical guarantee.
>
> 5. **On the $K_q$ (Q5).**
>
>    We define $K_q$ (L375 left) as a quantile-based sparsity statistic, with formal definition in L1592–1594 and implementation details in L1664–1671. In practice, for a set of representations, we count the number of activated units per sample, sort these counts, and take $K_q$ as the sparsity at the $q$-th quantile.
>
> 6. **On the Fig.5 (Q6).**
>
>    Each cell corresponds to an independently trained model. The x-axis denotes training data scale and the y-axis model scale, both normalized to a base of 9216 and scaled accordingly. For each $(x, y)$, the color shows the $R^2$ score, measuring reconstruction fidelity. Further details and motivation are in L358–379 (right) and the caption.
>
> 7. **On the Tab.1 (Q7).**
>
>    Tab.1 reports results averaged across all layers of each LLM, with full layer-wise results in Tab.3–5. All results are carefully verified and show that fundamental representational units are widely present, consistent with our theoretical expectations. We will clarify this and explicitly reference the full results in the revision.
>
> 8. **Response to W1&W2**.
>
>    We acknowledge the technical complexity of these symbols. To maintain a clear narrative in the Introduction, we defer detailed definitions to $\S$3–4. In the revision, we will add clear forward references in Fig.1 to help readers locate them.
>
> 9. **Response to W3**.
>
>    Please refer to our response to Reviewer VPEj (Point 5).
>
> 10. **Response to W5**.
>
>     We clarify the motivation behind this design. The radar chart reflects two core metrics: $R^2$ (faithfulness) and $q^* $ (stability). Ideal atoms satisfy $R^2 = 1$ and $q^* = 1$, corresponding to a fully filled unit circle. This circular form intuitively shows proximity to ideal atoms: fuller coverage indicates higher faithfulness and stability across layers. We will enlarge the figure and clarify the caption in the revision, including each dimension and the rationale for this visualization.
>
> 11. **Response to W6&W7**.
>
>     Our goal is not to claim neuron-based analysis is ineffective, but to note that superposition can hinder individual neurons from representing clear semantics, limiting the consistency and stability of interpretations based on single neurons. Moreover, we focus on the two main paradigms (neurons and features) as the primary objects in interpretability, which our work compares and unifies.
>
>     In the revision, we will clarify these points and briefly discuss other related directions, outlining their connections and differences with ours.
>
> 12. **Response to W8**.
>
>     Please refer to our response to Reviewer 1LTF (Point 1).
> We are happy to provide more detailed clarification if needed.

---

> > ### Author Rebuttal · Reviewer_iX87 · 2026-03-31
> >
> > One of my concerns was the lack of proper citations of prior works, and the authors still don't cite the relevant works, even in the rebuttal.
> > The authors promise to make multiple changes in the paper, but without being able to see the new version, I can not recommend accepting the paper in its current form.

---

> > > ### Author Response · Authors · 2026-04-01
> > >
> > > We sincerely thank the reviewer for the additional feedback. Due to the space limitations of rebuttal, our previous response was relatively brief and may not have fully clarified the situation, for which we apologize. We provide further clarification and elaboration below.
> > >
> > > **On the related work citation (W3).**
> > >
> > > - **On Definition 3.5 (coherence-related definition)**
> > >
> > >   Regarding Definition 3.5, it is introduced naturally within the context of \[7\] to characterize the structure of the problem we study. While this notion is indeed widely used in classical compressed sensing frameworks, the current manuscript only cites representative works \[1\]\[2\] in $\S$3.3. In the revision, we will provide systematic introduction to this concept and include relevant references \[5\].
> > >
> > > - **On the use of coherence in deep learning theory**
> > >
> > >   Our work does not follow existing lines of research that analyze deep networks directly through coherence. Instead, we introduce coherence as a foundational tool within a new problem setting (in connection with \[7\]\[8\]). We agree that this distinction is not sufficiently clarified in the current manuscript. In the revision, we will clearly demonstrate this point and include additional references \[6\] to strengthen the background.
> > >
> > > - **On Theorems 3.8 / 3.9**
> > >
> > >   We agree that these results are formally related to classical results in compressed sensing [3\]\[4\], while the current manuscript only cites representative works \[1\]\[2\] and does not sufficiently cover earlier or more directly related sources. In the revision, we will (i) explicitly clarify the relationship between these theorems and existing results; (ii) add appropriate citations [3\]\[4\]; and (iii) more clearly distinguish between classical results and our reorganization and derivation within a unified framework.
> > >
> > > We understand the reviewer’s concern regarding the inability to view the revised manuscript. Due to ICML policy, we are unable to update the PDF at this stage. However, we commit to fully incorporating the above revisions in the final version and clearly reflecting all additional citations and clarifications in the public release. We sincerely thank the reviewer again for the valuable feedback.
> > >
> > > [1] Compressed sensing, Donoho, David L.
> > >
> > > [2] Robust uncertainty principles: Exact signal reconstruction from highly incomplete frequency information, Candès, Emmanuel J., Justin Romberg, and Terence Tao.
> > >
> > > [3] Decoding by linear programming, Candes, Emmanuel J., and Terence Tao.
> > >
> > > [4] Uncertainty principles and ideal atomic decomposition, Donoho, David L., and Xiaoming Huo.
> > >
> > > [5] An invitation to compressive sensing, Foucart, Simon, and Holger Rauhut.
> > >
> > > [6] Can We Gain More from Orthogonality Regularizations in Training Deep CNNs?, Nitin Bansal et al.
> > >
> > > [7] Toy models of superposition, Nelson Elhage et al.
> > >
> > > [8] The linear representation hypothesis and the geometry of large language models, Park, Kiho, Yo Joong Choe, and Victor Veitch.

---

### Official Review · Reviewer_1LTF · 2026-03-12

**Soundness:** 3
**Presentation:** 3
**Significance:** 3
**Originality:** 3
**Overall Recommendation:** 4
**Confidence:** 4

**Summary:**

The paper investigates the fundamental representational units (FRUs)—termed "atoms"—within large language models (LLMs). The authors propose **"Atom Theory,"** introducing a non-Euclidean metric called the Atomic Inner Product (AIP) to correct a pervasive "representation shift" identified by the authors. Based on the AIP, the paper formally defines ideal atoms using quantitative criteria for representability/faithfulness ($R^2$) and stability ($q^*$). The authors theoretically prove and empirically demonstrate that threshold-activated sparse autoencoders (TSAEs, specifically using JumpReLU) can successfully identify these atoms, provided the autoencoder capacity rigorously matches the data scale. Extensive experiments across multiple LLM families (e.g., Gemma-2, Llama-3.1) show that the extracted atoms achieve near-perfect faithfulness and high stability, overcoming the well-documented limitations of traditional neurons and standard SAE features, ultimately yielding high monosemanticity.

**Compliance With Llm Reviewing Policy:**

Affirmed.

**Key Questions For Authors:**

See Weakness

**Limitations:**

See Weakness

**Strengths And Weaknesses:**

## Strengths

- S1 **(Significance & Scientific Contribution):** The quest to define and extract the foundational basic units inside LLM representations is a highly interesting and meaningful direction. By formalizing this problem through Atom Theory, the paper makes a solid scientific contribution.

- S2 **(Methodology & Resources):** The paper provides a comprehensive evaluation framework and rigorous methodology. The large-scale experimentation (spanning models like Gemma-2-9B and Llama-3.1-8B) and the systematic insights regarding the matching of TSAE capacity with data scale offer helpful resources and empirical heuristics for the interpretability community.

- S3 **(Theoretical Motivation):** The motivation behind the AIP theory is convincing to me. The transition from theoretical definition to empirical extraction provides strong evidence for the *existence* of these atomic structures in practice.

- S4 **(Representation Shift Finding):** The discovery of the "representation shift"—and the comprehensive real-world experiments demonstrating how the AIP metric correlates with and corrects this geometric distortion—is a particularly interesting and novel finding.

- S5 **(Effectiveness of TSAEs):** The theoretical justification and practical application of JumpReLU to find these atoms are well-founded, leading to valid findings that clearly justify the proposed method from standard ReLU-based SAEs.

## Weaknesses

- W1 **(Extensions & Applications):** The paper currently offers limited insights into how Atom Theory can be extended to the application side. It would be beneficial to add a dedicated discussion section exploring how these identified atoms could be practically utilized for AI safety, pure interpretability tasks, and active model steering.

- W2 **(Propagation of the Representation Shift):** The core claim that the *Softmax operation alters the geometry of LLM representations* (inducing the representation shift) is fascinating, but it is somewhat under-explored at the beginning of Section 3. The authors briefly state that this invariance propagates to all hidden representations due to the residual stream and the linearity of matrix multiplication. However, it remains unintuitive how this global bias propagates backwards through all layers given the multitude of complex non-linearities (e.g., SwiGLU, attention Softmax) inside the LLM. Further empirical or theoretical justification on this cross-layer propagation is needed to make this claim fully convincing.

- W3 **(Strong Assumption on Separability):** The theory relies heavily on a strong assumption regarding the separability of atoms (approximate orthogonality), which may not hold universally. A growing body of evidence suggests that language representations might not be clearly separable, as some features are high-dimensionally correlated, exhibit inherent multi-semanticity, or exist as semantic manifolds [1, 2, 3]. The current paper primarily focuses on word/entity-level interpretations (e.g., using the CounterFact dataset), where such linear separability is more likely to hold. The authors should discuss the limitations of this assumption for more abstract, complex cognitive representations like reasoning.

- W4 **(From Atoms to "Molecules"):** Building on the previous point, is it possible to use the discovered atoms as building blocks to form "molecules"? A discussion/experimentation on whether these mono-semantic atoms can be mathematically combined to reconstruct and explain the multi-semantic forms observed in higher-level LLM behaviors would significantly strengthen the paper's theoretical and practical scope.

References:

[1] Engels, J., Michaud, E. J., Liao, I., Gurnee, W., & Tegmark, M. Not all language model features are one-dimensionally linear. ICLR 2025.

[2] Modell, A., Rubin-Delanchy, P., & Whiteley, N. The origins of representation manifolds in large language models. arXiv preprint arXiv:2505.18235.

[3] Zhou, Y., Wang, Y., Yin, X., Zhou, S., & Zhang, A. R. The geometry of reasoning: Flowing logics in representation space. ICLR 2026.

---

> ### Author Rebuttal · Authors · 2026-03-30
>
> We sincerely thank you for your valuable feedback and evaluation. We respond to the comments as follows.
>
> 1. **On the suggestion to expand analysis and discuss applications (W1 & W4).**
>
>    We agree that exploring how the proposed atom theory can be extended for deeper analysis and applied to real-world scenarios is  important.
>
>    In the revision, we will add a discussion section about potential extensions and applications, including  but not limited to: **system behavior analysis**, where stable representational units serve as building blocks that can be systematically composed to characterize more complex semantic structures, thereby bridging low-level representations and high-level behaviors; **AI safety**, where identifying stable representational structures may improve cross-input consistency analysis and help detect anomalous behaviors or potential risk patterns; and **controlled intervention**, where stable units enable fine-grained modifications, such as knowledge editing or behavior steering. We will also distinguish between the aspects that are empirically validated in this work and those that remain future directions, avoiding overextending our current conclusions.
>
> 2. **On the propagation of representation shift (W2).**
>
>    Actually, LLMs contain various nonlinear components (e.g., SwiGLU, attention softmax), whose effects on the geometry of representations can be complex. In our current analysis, we indeed derive results based on the residual-stream architecture and linearity of matrix multiplication. This is an **intentional simplification**, aimed at **capturing the dominant mechanism of representation propagation**.
>
>    Specifically, our analysis focuses on how geometric shifts induced by the softmax operation are preserved and propagated along the residual stream, which serves as the primary pathway for information flow. Although nonlinear modules are present, they typically operate on local subspaces or modulate signals via gating mechanisms, which effect on the overall geometry empirically appears as **perturbations rather than complete restructuring**. At the experimental level, we observe that this shift remains consistent across different models and layers (see Fig 7–17), which provides empirical support for its stability in cross-layer propagation.
>
>    We acknowledge that the current manuscript does not sufficiently elaborate on this point. In the revision, we will (i) more explicitly clarify the simplifying assumptions underlying our theoretical analysis, and (ii) further discuss the potential impact of nonlinear components and the associated limitations.
>
> 3. **On separability and more general settings (W3).**
>
>    We agree that language representations often exhibit complex semantic structures, suggesting that semantics cannot, in general, be fully captured by a single linear direction. However, we would like to clarify that these observations are not in conflict with our formulation. For instance, in the related work [1], the original claim that “not all features are linear” has been further refined to “not all features are one-dimensional linear.” In other words, complex semantics can be represented by a set of basis directions rather than a single direction, which is also supported by their empirical findings.
>
>    From this perspective, our work focuses on a more fundamental level of structure: we aim to characterize the geometric properties of these basic representational units (atoms), rather than assuming that high-level semantics themselves are directly separable. In this sense, “approximate orthogonality” applies to these underlying units, enabling them to be combined in a way that ensures good distinguishability and stability, thereby supporting the composition of higher-level semantics.
>
>    Taken together, we view prior work on multi-semanticity and high-dimensional representations as **complementary rather than conflicting** with our perspective: such work describes the complexity of high-level semantics, whereas our work attempts to characterize their potential underlying structure.
>
>    In addition, to evaluate performance in more complex settings, we further conduct **task-level generalization experiments**. Specifically, we validate our method on **general corpora** (Wikipedia) to assess behavior under natural input distributions, and on **complex reasoning** (MATH500) to test its applicability in challenging reasoning scenarios. **The results (see L1525–1528, Table 2)** show that our method maintains consistent behavior patterns across these broader settings, suggesting that it extends beyond entity-centric knowledge scenarios and generalizes to more general LLM representation analysis.
>
> Due to space limitations, we are unable to elaborate further in this reply. We appreciate the reviewer’s understanding and would be happy to provide more detailed explanations if needed.
>
> [1] Not All Language Model Features Are One-Dimensionally Linear.

---

> > ### Author Rebuttal · Reviewer_1LTF · 2026-04-02
> >
> > Thanks for your rebuttal. Adding those discussions would make the paper better, especially for the simplification of assumptions in W2. This part needs to be modified as the overall argument still looks suspicious. Also, discussions on more related work of multi-semanticity and high-dimensional representations might be needed to distinguish the position of this paper. Others are good from my side. Thus, I will keep my score.

---

> > > ### Author Response · Authors · 2026-04-02
> > >
> > > Thank you for the valuable feedback and for acknowledging the improvements.
> > >
> > > We appreciate your suggestions regarding the clarification in W2 and the need for a clearer positioning with respect to multi-semanticity and high-dimensional representation literature. We agree that these aspects are important, and we will revise the manuscript accordingly to strengthen both the theoretical clarity and the discussion of related work.
> > >
> > > Thank you again for your thoughtful comments.

---

### Official Review · Reviewer_VPEj · 2026-03-15

**Soundness:** 3
**Presentation:** 3
**Significance:** 2
**Originality:** 2
**Overall Recommendation:** 3
**Confidence:** 2

**Summary:**

the paper proposes "atom theory" for defining and finding fundamental representational units inside large language models. the authors introduce a non-euclidean metric called atomic inner product that accounts for softmax invariance. atoms are defined through representability, sparsity, and separability. two criteria are proposed: faithfulness and stability. the paper proves that threshold-activated sparse autoencoders can identify atoms, and empirically shows that neurons and standard SAE features fall short while their method achieves near-perfect scores on several models.

**Compliance With Llm Reviewing Policy:**

Affirmed.

**Key Questions For Authors:**

1. the atomic inner product needs the atom set to be computed, but the atom set is what you are trying to find. in practice you estimate it differently. can you justify why this estimate is valid and when it breaks?
2. how do the identified atoms perform on general text activations, not just entity names? this is critical for claiming these are fundamental units of the model.
3 have you tried ReLU SAEs with larger capacity on the same data? maybe ReLU just needs more capacity rather than being fundamentally limited.
3 can you empirically verify that the conditions required by your identifiability theorem actually hold in practice?

**Limitations:**

the authors do not discuss key technical limitations: narrow training data, unclear generalization beyond entity activations, and the gap between theoretical assumptions and practical estimation.

**Strengths And Weaknesses:**

strengths:
1. the paper fills a real gap. the community uses neurons and SAE features without formal definitions of what makes a good representational unit. a rigorous framework for this is valuable.
2. connecting LLM representations to compressed sensing theory is natural and well-motivated. the insight that threshold activations are theoretically preferable over plain ReLU is useful.
3 the experimental scale is solid with three models and all layers covered.

weaknesses:
- the core theoretical results are mostly standard compressed sensing applied to a new setting. the coherence-RIP bound and uniqueness via mutual coherence are textbook material, see for example "a mathematical introduction to compressive sensing" by Foucart and Rauhut. the novelty is more in reframing than in new math.

- the atomic inner product is close to what Park et al. proposed in "the linear representation hypothesis and the geometry of large language models". the authors say theirs works on dynamic representations, but the practical difference is unclear. also in practice the metric is estimated as a whitening transform from random activations, which creates a gap between theory and experiments that is not well discussed.
- the experiments use a narrow data regime - entity-level activations from wikidata and counterfact, single token positions. it is unclear whether the found atoms generalize to typical LLM activations during normal use. if they only work for entity knowledge, calling them fundamental units of the whole model is a stretch.
- the comparison with gemma-scope and llama-scope is unfair. those are trained on broad corpora for general coverage, while the authors train on a controlled set of around 20K entities. getting higher reconstruction on your own training distribution is expected.
the monosemanticity evaluation uses an LLM judge with a simple counting prompt. this is a rough proxy and no large-scale human evaluation is provided.
- some layers had to be dropped due to optimization failure which raises robustness concerns.

---

> ### Author Rebuttal · Authors · 2026-03-30
>
> Sincerely thank you for your valuable feedback. We respond to the comments as follows.
>
> 1. **On the $\tilde S$ (Q1 & W2).**
>
>    **In principle, the AIP (on a broader level) is not restricted to “ideal atoms” defined in theory (the formulation in $\S$3.1 is introduced to facilitate the definition of atoms), but is broadly applicable to analyzing LLM representations and their potential linear decompositions (see L119–160 left).**
>
>    The $\tilde S$ used in our experiments is not meant to recover “ideal atoms,” but to serves as a **sample-based estimator of the underlying representational geometry**. Its construction varies slightly across settings:
>
>    - **Fig.3 / $\S$4.1.** This part aims to **validate the effectiveness of AIP**, rather than recover “ideal atoms.” Accordingly, we do not rely on a specific atom set, but construct directly from representations. Specifically, we collect representations from different layers across models to compute angle distributions. To estimate $\tilde S$, we sample representations $k$ from an **external , non-overlapping distribution** from Wikipedia, and compute $\tilde S = \mathbb{E}[kk^\top]$ (see L302–315 right).
>
>      The motivation is to estimate the inner product using independent samples, avoiding coupling with target representations. By linearity, this extends naturally to sub-units of representations (e.g., neurons, features, or ideal atoms) (see L130–159 left).
>
>    - **$\S$4.2–4.4.** We then focus on **specific representational units** (e.g., neurons or features), estimating the inner product under their induced distribution. For a set $D = \\{d_i\\}_{i=1}^{|D|}$, we compute $\tilde S = \mathbb{E}[dd^\top]$.
>
>      This follows “first selecting candidate units, then defining the corresponding metric” (see L191–193 right), aiming to evaluate them under an appropriate geometry. As independent, identically distributed samples outside the set are hard to obtain, we use in-distribution sampling. The estimator converges with sample size, making it empirically stable and well-justified.
>
>    We will clarify this in the revision.
>
> 2. **On more general settings (Q2 & W3).**
>
>    Our theory and analysis do not depend on how activations are obtained. In experiments, we use the current method for controllability and scalability (see L1519–1523). We also conduct task-level generalization experiments (see L1525–1528, Tab.2), including natural corpora and complex reasoning, showing consistent behavior across broader settings.
>
> 3. **On the capacity of ReLU SAE (Q2).**
>
>    We conduct controlled experiments with ReLU SAE under varying capacities, showing that increasing capacity does not improve performance.
>
>    | Capacity | 4x     | 5x     | 6x     | 7x     |
>    | -------- | ------ | ------ | ------ | ------ |
>    | $R^2$    | 0.9183 | 0.9173 | 0.9150 | 0.9143 |
>    | $q^*$    | 0.4768 | 0.4655 | 0.4514 | 0.4532 |
>
>    This is consistent with our theoretical expectation. We will include these results in the revision to support this conclusion.
>
> 4. **On the conditions of Theorem 3.11 (Q2).**
>
>    The condition in Theorem 3.11 is structurally consistent with $\mu < \frac{1}{2K - 1}$ in Theorem 3.9, implying that when the stability condition holds, sparse representations are unique and can be stably recovered via TSAE. In practice, this condition can be characterized statistically using $q^*$. We will clarify this connection in the revision to avoid misunderstandings.
>
> 5. **Response to W1**.
>
>    While some results relate to classical compressed sensing (see L205-210 right) and are used as tools, our contribution is not a direct application but lies in: (i) **Problem reformulation**: introducing compressed sensing to analyze LLM representations, turning FRUs discovery into a geometric problem; (ii) **Unified framework**: unifying definition, evaluation, and identification of FRUs with statistical measures; (iii) **Targeted derivation**: constructing theorems under our setting using minimal assumptions. We will clarify this in the revision.
>
> 6. **Response to W2.**
>
>    Park et al. analyze the unembedding matrix (on output tokens) using an inner product defined in a fixed, input-independent parameter space (see their $\S$3.1 and official implementation in store_matrices.py ). In contrast, we study an input-dependent representation space, defining AIP to characterize geometry among representations and their constituent units.
>
> 7. **Response to W4.**
>
>    In Fig.2, we show that neurons and features have a substantial gap from "ideal atoms" in faithfulness and stability. In Tab.1, rather than direct comparison, we demonstrate that our units are closer to "ideal atoms" on both metrics. We will clarify the distinction between comparative analysis and fair benchmarking in the revision.
>
> 8. **Response to W5.**
>
>    We later identify this issue as due to higher representation variance in these layers. We have addressed it by adjusting the training setup and will clarify this in the revision.

---

### Official Review · Reviewer_TtT4 · 2026-03-16

**Soundness:** 3
**Presentation:** 3
**Significance:** 2
**Originality:** 2
**Overall Recommendation:** 5
**Confidence:** 2

**Summary:**

The authors propose a definition of “atoms” for LLM representations — a minimal set of vector-valued features which are approximately orthogonal and sparsely activated for any given input, and which jointly compress hidden LLM activations. They provide theoretical arguments to justify their definition, prove identifiability of these atoms under certain assumptions, and provide a simple method for extracting atoms from LLM activations using thresholded sparse auto-encoders. Finally, they show that this sparse auto-encoder method discovers atoms which reconstruct the hidden activations with high fidelity, remain sparsely activated, and remain approximately orthogonal. Empirically, the discovered atoms appear highly “monosemantic” — i.e., they represent a single underlying “concept” or “meaning”.

**Compliance With Llm Reviewing Policy:**

Affirmed.

**Final Justification:**

In the end, I stand by my initial score. I had a positive impression of the paper that has remained after reading the other reviews and engaging with the authors. Most of the concerns that I did have were addressed by the authors, and the final point that remained unaddressed (lacking experiments with real-world applications) aren't significant enough to reduce my score. I believe that the paper is sound and is a meaningful contribution to the field.

**Key Questions For Authors:**

##

1. For Section 4.2, what are “features”? I think the main text should make this clear without needing to refer to the Appendix, given that the term is not well-defined and understanding what exactly you used is crucial to understanding how/why features are poor candidates for atoms.
2. In Section 4.3, how is model capacity scaled? Are we talking about encoder/decoder capacity, or are we talking about hidden state dimensionality (i.e., maximum number of atoms)?
3. Regarding Section 4.3 again, why can’t we just always overparameterize the TSAE? Why do we care about making it only as large as it needs to be for the amount of data we have? Does stability suffer if it is given too much capacity?
4. Other than saying which candidate atom sets are along the pareto front, how are we to do model selection empirically?
5. Why the limited scope and emphasis on natural language and LLMs? Is there any reason in particular why this can’t be used as a generic interpretability method for deep neural network representations in arbitrary data modalities and neural architectures (or even for biological neural data)? Are any of the assumptions and methods specific to natural language and LLMs?
6. What can the resulting atoms be used for? What problems in AI do they solve? Can you demonstrate such applications empirically?

**Limitations:**

Limitations are not discussed. It would be nice to include a discussion around this. Clearly in terms of “negative societal impact”, the work is likely a net positive (as the authors write in their impact statement), but surely you can think of technical limitations and some negative social consequences. For instance, bad actors might love to extract these atoms so that they can do surgery on an LLMs internal state to drive it towards whatever goal they might have, whereas controlling a black box is more difficult in comparison. And, if these atoms rely on assumptions that don’t hold in practice, they might only provide a false sense of comfort and understanding for a system that in reality is operating in completely different (and inscrutable) ways.

**Strengths And Weaknesses:**

## Strengths

1. Sparsity is introduced in the definition as a soft criteria (i.e., a score) for a set of atoms, rather than a constraint with an arbitrarily-defined sparsity level.
2. More often than not, following an abstract mathematical definition or a series of derivations, a short paragraph is dedicated to providing concrete intuition. This greatly helps with understanding without making the paper too verbose.
3. The proposed method seems simple enough to apply in practice, and as far as I can tell doesn’t rely on too many hyper-parameters. This should help with wide adoption in the community if the proposed method does become the standard.
4. There’s significant interest in sparse auto-encoders for finding interpretable concepts in LLMs. The proposed method seems to build on this work, and as far as I can tell strictly improves on it. So it’s easy for me to see how this is a meaningful step forward for the interpretability community.

## Weaknesses

1. Some of the desiderata for atoms seem unjustified. Why do they need to be approximately orthogonal? Why do they need to be sparse? Are we just trying to compress the representation here, or are we trying to uncover interpretable models of LLM representations? If it’s the former, then I think the desiderata are perhaps easy to justify. If it’s the latter, then I think the connection is too loose for me at the moment. Certainly if I draw the analogy between atoms and “mental concepts”, “mental concepts” don’t seem to be orthogonal and independent; in fact they are deeply inter-dependent and have rich relational structure. I realize that the stated desiderata are quite standard in the interpretability community, but I’ve always found these assumptions questionable and I think the paper should make an effort to justify them without taking them as granted and self-evident.
2. Section 3.1 comes quite out of nowhere with little context (at least to a reader like me not deeply immersed in the literature around LLM interpretability). Prior to this section, all that was said about the atomic inner product (AIP) was that it is some sort of correction that takes into consideration the softmax operation of LLMs. It isn’t clear to me by this point in the paper why the AIP is being introduced and why it is necessary to the definition of an atom, so these really feel like definitions and theorems without any context. Perhaps something more conceptual needs to be added to this section in order to prepare the reader for *why* the AIP is going to be important later on.
3. The same point can be made for Sections 3.2 and 3.3. I find myself not really grasping the high-level direction by this point in the paper, so the mathematical derivations are coming out of context. It would be helpful to preempt Section 3 as a whole with some conceptual outline that clarifies why the mathematical steps will unfold the way they do.
4. The claims of identifiability early in the paper should come with a caveat, because, as is stated later in the paper, they assume that the representation is formed as a $K$-sparse linear combination of ground-truth atoms. While it’s nice that under this assumption the atoms can be recovered, the assumption itself is doing a lot of the work here. Unless I misunderstood something, I’m sure the authors would agree that this assumption is only likely to be *approximately* true (at best) for real LLMs, and the real question is about how sensitive the approach is to deviations from this assumption. In other words, when representability is not 1.0, what value does the identifiability result still provide?
5. The authors discuss monosemanticity as more of an aside, but it seems to me that monosemanticity is isn’t rigorously defined in the first place (what is a “concept” or “meaning” in this case?), and it’s evaluation is rather ad-hoc and subjective. Why not just remove this discussion altogether?
6. Section 4.3 seems tangential to the central messages of this paper, and the conclusions (”Intuitively, the data scale determines the scale of FRUs…”) seem obvious. I would recommend cutting it and just considering overparameterized TSAEs.
7. Scoring atoms involves considering two metrics, faithfulness and stability. It seems like we can always sacrifice one to improve the other (fewer sparsity constraints → better reconstruction). It therefore isn’t clear how to judge which candidate atom sets are better or worse if they both lie along the pareto front.
8. Maybe I missed something, but the baselines seem rather weak. Of course individual neurons are going to fail along the stability score given their lack of sparsity, so any sparsity-based construction method will trivially result in better stability. From the main paper, I didn’t follow what “features” corresponded to (see my question below), so perhaps that baseline is better and addresses my concerns. But ultimately, there are numerous methods that try to find “atoms” or “concepts” underlying deep neural network representations. How does the proposed method relate to these alternatives?
9. The results presented in the main text (i.e., ignoring the Appendix) seem quite shallow. The authors proposed two metrics for atoms which their method is explicitly designed to optimize for, and then show that it maximizes these metrics better than competing methods. That conclusion seems obvious. But ultimately I think the current approach to presenting the results fails to address the questions of: (1) why do we care about atoms in the first place / what problems does this help us solve, (2) does the proposed method in fact solve these problems better than alternatives? Right now, all I know is that the proposed metric discovers atoms that an LLM judge thinks are more “monosemantic” than competing methods. What do we as a community do with this information?
10. Minor: a few grammar typos in Section 3.4 that are worth correcting (e.g., “This respond to O’Neill…”, JumpReLU is introduced to addresses feature shrinkage…”).

---

> ### Author Rebuttal · Authors · 2026-03-30
>
> Sincerely thank you for your valuable feedback and evaluation. We respond to the comments as follows.
>
> 1. **On the definition of features (Q1&W8).**
>
>    In existing literature, “features” lack a universal formal definition and are typically treated operationally as directions in representation space associated with certain semantic patterns. Following this convention, we treat directional units from existing tools (e.g., GemmaScope, LlamaScope) as features, using them as candidate units for comparison with neurons and ideal atoms. We will clarify this in the revision to avoid potential ambiguity.
>
> 2. **On the experiments in §4.3 (Q2&Q3&W6).**
>
>    By “model capacity,” we refer to the encoder/decoder capacity (i.e., dictionary size or hidden units), not the representation dimensionality. It determines the maximum number of learnable representational units and can be scaled via model configuration.
>
>    Overparameterization is feasible and supported by our experiments: excess capacity remains unused, with the model relying on effectively activated units. In this case, stability is determined by these active units and typically does not degrade with increased capacity. However, our  primary consideration here is computational efficiency, as excessive capacity increases training cost without yielding additional benefits.
>
> 3. **On the faithfulness and stability, and model selection (Q4&W7).**
>
>    We agree these metrics may trade off in principle, but empirically this is not actually the case. In early and mid training, faithfulness and stability improve jointly, reflecting more structured representations. A mild trade-off appears only in later stages when sparsity is further enforced.
>
>    Importantly, under our setup, we typically obtain units with both high faithfulness and stability, suggesting they can be jointly optimized. We will add discussion of training dynamics (Fig.37) in the revision to clarify this relationship.
>
> 4. **On the applicability of the theoretical framework (Q5).**
>
>    Our framework does not depend on natural language or specific LLM architectures. We study language models for their rich representational structures and extensive interpretability work, facilitating systematic analysis. Methodologically, core components (e.g., AIP) arise from gauge symmetry in representation space, driven by linear transformations common across neural networks, not specific to LLMs. Thus, the framework can extend to other architectures and modalities in principle. We will clarify which aspects are architecture-agnostic vs. design-dependent in the revision. Preliminary experiments on other models (e.g., diffusion) show similar behavior, supporting generality, though detailed analysis is beyond scope.
>
> 5. **On extended analysis and application discussion (Q6&W9).**
>
>    Please refer to our response to Reviewer 1LTF (Point 1).
>
> 6. **On the motivation and justifiability of the theoretical framework.**
>
>    - **Why is the AIP needed ($\S$3.1)? (W2)**
>
>      In language models, information is encoded in high-dimensional representations. Analyzing their geometry requires specifying an inner product, as it defines the underlying geometry. Thus, in $\S$3.1, we establish an inner product aligned with representation structure, forming the basis for subsequent analysis. We will clarify this in the revision.
>
>    - **Why “approximate orthogonality + sparsity”? (W1)**
>
>      The core intuition is to identify fundamental representational units (atoms) that are detectable or manipulable without interfering with others (distinguishability) (see L129-135 left). Ideally, this corresponds to strict orthogonality, but in an $H$-dimensional space, at most $H$ such vectors exist, which limits expressiveness.
>
>      Compressed sensing shows strict orthogonality is unnecessary, and distinguishability can be preserved via sparsity and approximate orthogonality. **Thus, this combination is not an added assumption but a mathematical generalization of distinguishability (see L208–214 left; L183–189 right)**.
>
>    - **Relationship among definition, evaluation, and identifiability (W3).**
>
>      We introduce the definition of atoms ($\S$3.2), evaluation metrics ($\S$3.3), and identifiability with theoretical guarantees ($\S$3.4). These components are developed progressively around the central question of FRUs (atoms).
>
>    - **On the approximate condition (W4).**
>
>      We agree this condition may be only approximately satisfied (representability is not 1). Rather than assuming they hold exactly, we quantify their approximation using metrics $R^2$ and $q^*$. Thus, even when representability is not 1, the theory remains meaningful as an interpretable reference for assessing how closely we approach a faithful and stable structure.
>
> We sincerely thank the reviewer for detailed and constructive feedback again. Due to space limitations, some responses are concise. We would be happy to provide further clarification and deeper analysis if needed.

---

> > ### Author Rebuttal · Reviewer_TtT4 · 2026-04-01
> >
> > Thanks for your rebuttal. I’ll focus my response on the more significant points from my initial review that I think remain unaddressed.
> >
> > >**3. On the faithfulness and stability, and model selection (Q4&W7).**
> > >
> > > We agree these metrics may trade off in principle, but empirically this is not actually the case. In early and mid training, faithfulness and stability improve jointly, reflecting more structured representations. A mild trade-off appears only in later stages when sparsity is further enforced.
> > >
> > > Importantly, under our setup, we typically obtain units with both high faithfulness and stability, suggesting they can be jointly optimized. We will add discussion of training dynamics (Fig.37) in the revision to clarify this relationship.
> > >
> >
> > I’m not saying both can’t be optimized jointly, and I’m not making a point about training dynamics. What I’m saying is that it seems subjective which candidate set of atoms along the pareto front you would say are “the best” or “the true” atoms. For instance, say that for one set of candidate atoms $D_A$ we have that that representability $R_A^2 = x$ and stability $q_A^* = y$, and for a second set of candidate atoms $D_B$ (perhaps obtained from training your TSAE with different hyperparameters) we obtain a different set of candidate atoms $D_B$ with representability $R_B^2 = x + \alpha$ and stability $q_B^* = y - \beta$, with $\alpha>0, \beta>0$. In this case, which is the “better” candidate atom set: $D_A$ or $D_B$? If you have an answer to this, please let me know. **If the selection between these two candidates (or any candidates along the pareto front) is subjective, then this undermines the work because it would mean that in real models there is no “true” candidate set of atoms.** If I’ve misunderstood something in your work and the point I’m making doesn’t apply, I apologize.
> >
> > > **5. On extended analysis and application discussion (Q6&W9).**
> > >
> > > Please refer to our response to Reviewer 1LTF (Point 1).
> > >
> >
> > Adding a discussion about how your method can be applied to real-world problems is helpful, but actually demonstrating these applications empirically would make for a stronger paper.
> >
> > > **On the approximate condition (W4).**
> > >
> > > We agree this condition may be only approximately satisfied (representability is not 1). Rather than assuming they hold exactly, we quantify their approximation using metrics $R^2$ and $q^*$. Thus, even when representability is not 1, the theory remains meaningful as an interpretable reference for assessing how closely we approach a faithful and stable structure.
> > >
> >
> > I understand that you discuss these $R^2$ and $q^*$ metrics for cases in which the assumptions backing up your identifiability result don’t hold. My comment was more along the lines of: what’s the purpose of your identifiability result if the assumptions behind it will *never* hold in practice? Why include the identifiability result at all, rather than just discuss scoring along your two empirical metrics? I don’t understand why the identifiability result is useful, unless it comes with some kinds of theoretical bounds about what should happen when the (strong) assumptions it makes are violated.

---

> > > ### Author Response · Authors · 2026-04-02
> > >
> > > We sincerely thank the reviewer for the response. Due to rebuttal length limits, some replies were necessarily concise. We provide additional clarifications below.
> > >
> > > 1. **On the faithfulness and stability, and model selection (Q4&W7).**
> > >
> > >    We understand the reviewer’s concern: when multiple candidate solutions lie on the Pareto frontier, is there a principled criterion to select the “true” set of atoms?
> > >
> > >    We clarify this as follows:
> > >
> > >    - From a theoretical perspective, “ideal atoms” correspond to structures that simultaneously achieve high faithfulness and stability ($R^2 = 1$ and $q^* = 1$). In practice, due to optimization limitations and finite data, this ideal point cannot be exactly attained, but can be approached. As a result, candidate solutions on the Pareto frontier typically lie in a near-ideal regime (see Figure 37). Within this regime, differences among candidates are often minimal, and the resulting representational structures are functionally similar. Therefore, the selection is not “arbitrary subjective,” but rather akin to choosing within an approximate equivalence class that satisfies the theoretical criteria.
> > >    - In practice, an operational selection criterion is still required. In this work, we adopt a simple and stable rule (balanced optimal of the two metrics; see L1558–1560). More importantly, our empirical observations show that, within a reasonable training range, different selection strategies (e.g., early stopping, different weightings, or hyperparameters) tend to converge to structurally consistent solutions (see Figure 38). Thus, the existence of a Pareto frontier does not weaken the significance of the problem; rather, it indicates the presence of a class of structurally constrained stable solutions, instead of a single exact point.
> > >    - Compared to prior work (which often relies on human judgment to assess the monosemanticity of representational units), our contribution is to transform this problem from a subjective evaluation into a structure-constrained selection problem based on computable metrics. Even though some degree of freedom remains within the Pareto frontier, it is significantly restricted to a region with clear theoretical meaning. We will clarify this point more explicitly.
> > >
> > > 2. **On extended analysis and application discussion (Q6&W9).**
> > >
> > >    We agree that further validation through concrete empirical applications would strengthen its impact.
> > >
> > >    In this work, our primary objective is to establish a theoretical framework for characterizing, evaluating and identifying FRUs in representations. Accordingly, the paper focuses on: (i) definition and theoretical characterization; (ii) the development of evaluation metrics; and (iii) systematic validation across models, rather than optimization for specific downstream applications. Moreover, we take initial steps to assess its generalization and practical potential. Experiments on general corpora (Wikipedia) and complex reasoning (MATH500) show consistent behavior across data distributions and task complexities (see Table 2), providing preliminary support for real-world applicability.
> > >
> > >    We agree that concrete applications (e.g., interpretability, intervention, safety) is an important next step. Due to space constraints and scope, we do not explore these. However, we will include additional discussion and plan to systematically investigate them in future work. We sincerely thank the reviewer for this valuable suggestion.
> > >
> > > 3. **On the approximate condition (W4).**
> > >
> > >    We understand this concern: if the conditions required by the identifiability theorem do not strictly hold in practice, what is the role of the theorem itself?
> > >
> > >    We clarify that Theorem 3.11 is not intended to assume that these conditions are fully satisfied in real systems, but rather to characterize an ideal target regime, under which representational units are uniquely identifiable. Building on this, we use $q^* $ to quantify the extent to which a real system deviates from this ideal regime. In other words, the theorem provides a structural reference, while $q^* $ measures how closely the system approaches it. These two components are therefore complementary rather than interchangeable. If we rely solely on empirical metrics, we lack a clear theoretical interpretation. The identifiability theorem provides this interpretation: when the system approaches the stated conditions, we can expect its representations to exhibit stable and unique structure.
> > >
> > >    In this sense, the value of the identifiability result lies in linking empirical metrics to theoretical properties, transforming them from heuristic scores into quantities with clear structural meaning. We will more explicitly clarify (i) the role of the theorem as a structural target; (ii) how the metrics capture this condition; and (iii) how deviations from this condition affect representation structure. We sincerely thank the reviewer for this insightful question.

---

### Decision · Program_Chairs · 2026-04-30

**Decision:**

Accept (regular)

**Comment:**

This paper addresses how to incorporate new knowledge into pretrained language models without degrading existing capabilities. It proposes a framework that models next-token prediction as a Markov process over tokens and introduces new knowledge via a token-to-dictionary mapping learned through lightweight embedding updates. Reviewers agreed that this perspective is intuitive and novel, and that the approach offers an appealing path toward parameter-efficient adaptation with minimal forgetting.

Across reviews, the main strengths are the originality of the Markov formulation, the accompanying sample complexity analysis that provides theoretical grounding for embedding-only updates, and the empirical results showing strong retention of prior knowledge. While the initial experiments were mostly synthetic, the additional results provided during the rebuttal, particularly on cross-lingual transfer, including more distant language pairs, strengthened the evidence that the method can be effective in more realistic settings.

The primary concerns focus on the gap between theory and practice. The theoretical guarantees rely on simplifying assumptions (e.g., restricted transition patterns) that do not fully reflect real generation behavior, and the extensions to more general settings are only sketched rather than formally developed. In addition, the experimental section in the original submission is somewhat limited in scope, and the positioning with respect to related work (e.g., vocabulary expansion and cross-lingual transfer) could be clearer. These concerns were raised by multiple reviewers and partially addressed in the rebuttal.

The rebuttal addressed several of the key concerns in a substantive way. The authors clarified that the restrictive assumptions in the theory are primarily for analytical tractability and are not imposed in practice, and they provided a more detailed explanation of how the framework can extend to more general transition settings. In addition, they introduced new experiments, covering cross-lingual transfer across multiple models and including more distant language pairs, that directly test the method in less controlled and more realistic scenarios. These results provide additional evidence that the approach can generalize beyond the synthetic setups in the original submission and help narrow the gap between theory and practice. The authors also committed to improving the paper’s positioning and clarifying the scope of their contribution in the final version.

After carefully considering the reviews, author responses, and discussion, I recommend acceptance. The core contributions, a principled and novel formulation of LLM adaptation together with meaningful theoretical insights, is technically sound and non-trivial. Importantly, reviewers who were initially uncertain were positively influenced by the rebuttal, and I have also taken the authors’ clarifications and additional experimental evidence into account. While some limitations remain, they are largely a matter of scope and presentation rather than fundamental flaws.

For the final version, the authors should include all edits from the discussion in particular:
- clearly separate proven results from conjectural extensions beyond the simplifying assumptions.
- integrate the additional experiments from the rebuttal with sufficient detail.
- improve the positioning and discussion of practical applicability and limitations, including and discussing the following references as indicated by reviewers:
  -   Deep Neural Networks with Random Gaussian Weights: A Universal Classification Strategy?
  - Convolutional neural networks analyzed via convolutional sparse coding
  - Dataless Model Selection with the Deep Frame Potential
  - An ETF view of Dropout regularization
  - The Restricted Isometry of ReLU Networks: Generalization through Norm Concentration
  - Conditioning of random Fourier feature matrices: double descent and generalization error
  - Overparameterized ReLU Neural Networks Learn the Simplest Model: Neural Isometry and Phase Transitions
  - Provable Identifiability of Two-Layer ReLU Neural Networks via LASSO Regularization